# Caspase-11 promotes allergic airway inflammation

Zbigniew Zasłona[1✉], Ewelina Flis[1], Mieszko M. Wilk [1], Richard G. Carroll [1], Eva M. Palsson-McDermott[1], Mark M. Hughes[1], Ciana Diskin[1], Kathy Banahan[1], Dylan G. Ryan[1], Alexander Hooftman [1], Alicja Misiak[1], Jay Kearney [1], Gunter Lochnit [2], Wilhelm Bertrams [3], Timm Greulich[4], Bernd Schmeck[3,4], Oliver J. McElvaney[5], Kingston H.G. Mills [1], Ed C. Lavelle [1], Małgorzata Wygrecka [2], Emma M. Creagh [1] & Luke A.J. O'Neill [1]

Activated caspase-1 and caspase-11 induce inflammatory cell death in a process termed pyroptosis. Here we show that Prostaglandin $E_2$ ($PGE_2$) inhibits caspase-11-dependent pyroptosis in murine and human macrophages. $PGE_2$ suppreses caspase-11 expression in murine and human macrophages and in the airways of mice with allergic inflammation. Remarkably, caspase-11-deficient mice are strongly resistant to developing experimental allergic airway inflammation, where $PGE_2$ is known to be protective. Expression of caspase-11 is elevated in the lung of wild type mice with allergic airway inflammation. Blocking $PGE_2$ production with indomethacin enhances, whereas the prostaglandin $E_1$ analog misoprostol inhibits lung caspase-11 expression. Finally, alveolar macrophages from asthma patients exhibit increased expression of caspase-4, a human homologue of caspase-11. Our findings identify $PGE_2$ as a negative regulator of caspase-11-driven pyroptosis and implicate caspase-4/11 as a critical contributor to allergic airway inflammation, with implications for pathophysiology of asthma.

[1] School of Biochemistry and Immunology, Trinity Biomedical Sciences Institute (TBSI)Trinity College Dublin, Dublin, Ireland. [2] Department of Biochemistry, Faculty of Medicine, Justus Liebig University, Giessen, Germany. [3] Institute for Lung Research, Universities of Giessen and Marburg Lung Center, Philipps-University Marburg, Member of the German Center for Lung Research (DZL), Marburg, Germany. [4] Department of Medicine, Pulmonary and Critical Care Medicine, University Medical Center Giessen and Marburg, Philipps-University, Member of the German Center for Lung Research (DZL), Marburg, Germany. [5] Royal College of Surgeons in Ireland, Beaumont Hospital, Dublin, Ireland. ✉email: zaslonaz@tcd.ie

Cell death is an integral component of the inflammatory process. Several different types of cell death have been identified that can influence the inflammatory response by causing tissue damage and contribute to, or even mediate pathology in a number of inflammatory diseases[1,2]. Caspase-11 (or its human homologs caspase-4 and -5) induces a type of cell death called pyroptosis, which is highly inflammatory and promotes tissue injury[3]. Caspase-11 is activated by the direct binding of cytosolic lipopolysaccharide (LPS) from Gram-negative bacteria, and is an essential mediator of sepsis[4,5], but can also be activated by endogenous signals from dying cells[6]. The role of caspase-11 beyond bacterial infections has not been extensively explored, and very little is known about the negative regulation of caspase-11.

PGE$_2$ was recently shown to inhibit the NLRP3 inflammasome, a key driver of IL-1$\beta$ production and pyroptosis[7,8]. PGE$_2$ exhibits versatile pro-inflammatory activities, such as increasing vascular permeability, generation of fever, and pain[9]. The importance of the pro-inflammatory actions of PGE$_2$ is highlighted by the wide clinical use of nonsteroidal anti-inflammatory drugs (NSAIDs), such as aspirin or indomethacin—that block PGE$_2$ production to relieve inflammation. Paradoxically, PGE$_2$ also contributes to the resolution phase of inflammation, facilitating tissue regeneration, and a return to homeostasis[10,11]. Particularly, in asthma PGE$_2$ has been demonstrated to be protective, first inhaled[12], and later endogenous PGE$_2$[13] inhibited bronchoconstriction in asthmatics. The importance of the protective nature of PGE$_2$ in asthma is highlighted by a specific clinical subtype of asthma termed aspirin-induced asthma[14]. Exogenous PGE$_2$ was protective in the population of patients whose asthma is triggered by NSAIDs[15]. Although the effect of PGE$_2$ in inflammation is somewhat controversial[16], specifically in asthma it is not only protective by inhibition of bronchoconstriction but also by inhibition of inflammation[17,18]. The protective effects of PGE$_2$ have never been attributed to the inhibition of caspase-11-driven pyroptosis, to the best of our knowledge. Given the potential of caspase-11 in inflammation and tissue injury, we have explored the effect of PGE$_2$ on caspase-11 induction in the context of asthma.

We have found that PGE$_2$ inhibits induction of caspase-11 expression by suppressing type I interferon production, and that this may have particular relevance to asthma. PGE$_2$ is known to be protective in asthma[17,19] and the murine model of allergic airway inflammation[20]. GWAS studies identified a strong association with the gene encoding Gasdermin B[21], which promotes caspase-4-mediated pyroptosis[22] and asthma pathogenesis[23]. The role of caspase-11/4 and the process of pyroptosis have not been explored in asthma. We implicate caspase-11 in asthma using the ovalbumin model of allergic airway inflammation, which has features similar to asthma, and confirm the negative effect of PGE$_2$ on caspase-11 in the model, providing a putative explanation for why NSAIDs, which block prostaglandin production, might exacerbate asthma. These findings also further validate the targeting of caspase-11 and pyroptosis by PGE$_2$. We therefore identify a mechanism for caspase-11 inhibition via PGE$_2$, which could be important for the anti-inflammatory and tissue-protective effect of PGE$_2$, with a particular relevance in asthma pathogenesis.

## Results
**Prostaglandin E$_2$ protects against pyroptosis.** We first explored whether PGE$_2$ could inhibit caspase-11-mediated pyroptosis. To induce pyroptosis, we have used an established model of macrophage transfection with LPS encapsulated in liposomes. PGE$_2$ inhibits macrophage function by signaling through protein kinase A (PKA)[24,25]. Pretreatment with PGE$_2$ or with the PKA agonist

(N$^6$-benzyladenosine-3′,5′-cyclic monophosphate) decreased pyroptosis in BMDMs transfected with LPS (Fig. 1a). We sought to determine how PGE$_2$ might regulate caspase-11 and pyroptosis. Pretreatment of LPS-activated BMDMs with PGE$_2$ was found to inhibit caspase-11 transcription (Fig. 1b), resulting in inhibition of caspase-11 protein expression (Fig. 1c). PGE$_2$ acts by binding four types of receptor EP1–EP4. The EP2 receptor is responsible for inhibiting macrophage function[24] and plays a role in asthma[20]. The preliminary data show that EP2 receptor-deficient BMDMs had increased expression of caspase-11, suggesting that endogenous PGE$_2$–EP2 signaling is an inhibitory signal on caspase-11 expression in resting macrophages (Fig. 1d). This indicates that the inhibitory effect of PGE$_2$ on pyroptosis could be via inhibition of expression of caspase-11.

**Prostaglandin E$_2$ inhibits caspase-11 expression.** Next, we examined the mechanism of transcriptional inhibition of caspase-11. In LPS-treated BMDMs, IFN-$\beta$ signaling induces caspase-11 transcription through STAT-1 activation[26,27]. Consistent with a previous report[28], PGE$_2$ blocked LPS-induced IFN-$\beta$ production (Fig. 2a). PGE$_2$ was unable to inhibit caspase-11 expression induced directly by IFN-$\beta$ (Fig. 2b, compare lanes 3–6). This demonstrated that inhibition of LPS-induced IFN-$\beta$ production by PGE$_2$ is responsible for the reduced caspase-11 levels. Inhibition of IFN-$\beta$ production by PGE$_2$ treatment attenuated phosphorylation of STAT-1 (Fig. 2c, compare lane 4 to lane 3 for 2 h), and preliminary data show decreased binding of STAT-1 to the caspase-11 promoter (Fig. 2d). The transcriptional inhibition of caspase-11 expression by PGE$_2$ is depicted in Fig. 2e.

**Prostaglandin E$_2$ inhibits capase-11- and caspase-4-driven pyroptosis.** We next explored if PGE$_2$ would inhibit pyroptosis following the induction of caspase-11. When PGE$_2$ was added after LPS priming and 30 min before LPS transfection, pyroptosis was significantly inhibited (Fig. 3a), with unchanged caspase-11 protein expression (Fig. 3b, compare lane 3 to lane 4). The lack of effect on caspase-11 protein expression indicated that PGE$_2$ is unlikely to be inhibiting ongoing TLR4 signaling here; however, we cannot exclude that PGE$_2$ affects caspase-11 expression after LPS was transfected.

To verify our findings in humans, we used human macrophages differentiated from PBMCs from healthy volunteers, and found that PGE$_2$ decreased caspase-4 expression (Fig. 3c). PGE$_2$ given before or after priming inhibited pyroptosis in human monocyte-derived macrophages transfected with LPS (Fig. 3d). Since caspase-11 is a receptor for both LPS and oxidized phospholipids[6], we next explored if PGE$_2$ or the PGE$_2$ precursor, arachidonic acid (AA), which is present in phospholipids, could directly interact with caspase-11. We have confirmed that caspase-11 is indeed binding to LPS; however, neither PGE$_2$ nor AA was able to directly bind to LPS (Fig. 3e). This shows that PGE$_2$ is unlikely to directly interfere with the process of caspase-11 binding to LPS. Therefore, future work should address the mechanism of PGE$_2$-mediated inhibition of caspase-11 activity.

**Caspase-11 expression is induced in allergic lung inflammation.** We next explored an in vivo model where PGE$_2$ is known to be protective. Asthma is a disease where PGE$_2$ was demonstrated to be protective in humans[17,19] and in animal models[29,30]. In the OVA mouse model of allergic airway inflammation (AAI), we observed a dramatic increase in caspase-11 expression (Fig. 4a, compare lanes 1–4 to 5–8, quantified in Fig. 4b). Treatment of mice with exogenous PGE$_1$ analog in the form of misoprostol before sensitization, and each airway challenge, which as we have shown previously attenuates asthma[20], resulted in decreased total

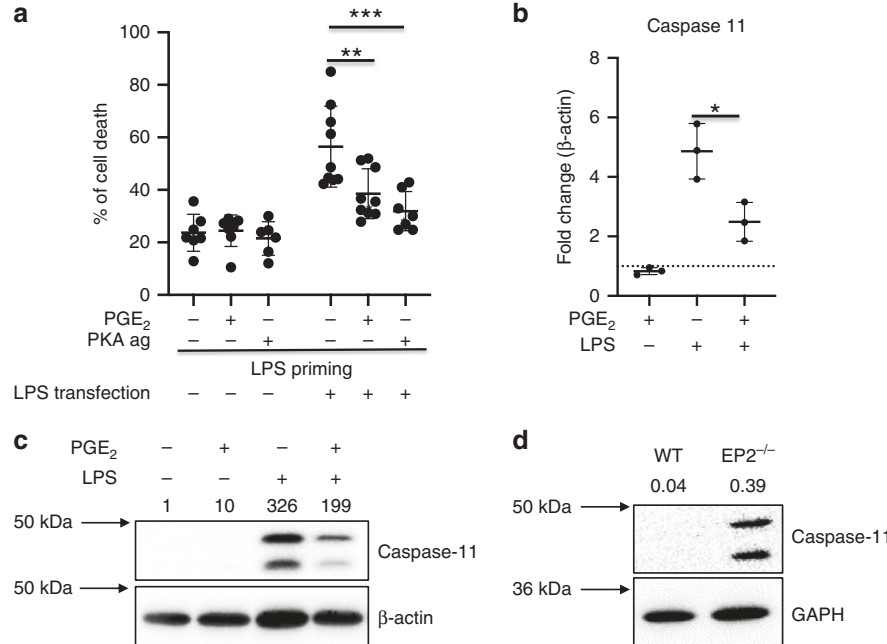

**Fig. 1 Prostaglandin E2 protects against pyroptosis. a** Murine BMDMs were treated with 1 μM $PGE_2$ or 500 μM 6-Bnz-cAMP (PKA ag) for 30 min followed by priming with 100 ng/ml of LPS for 4 h and 2 μg of LPS transfection using FuGENE 0.25% v/v liposomes o/n. Supernatants were collected and analyzed for cell death using an LDH assay. The results shown are from three independent experiments (with 2–4 mice per group in each experiment. Individual data points on the graph represent means from two to three technical replicates). Ordinary one-way ANOVA with Tukey's multiple comparison test has been used, **$P < 0.01$, ***$P < 0.001$, error bars represent mean ± SD. **b** BMDMs were treated with 1 μM $PGE_2$ or DMSO for 30 min followed by 100 ng/ml LPS for 4 h, and subjected to qPCR analysis of caspase-11 expression. The results shown are from a single experiment, with three mice in each group, and are representative of three independent experiments; individual data points on the graph are means from technical duplicates. *$P < 0.01$, two-tailed Student's *t* test, error bars represent mean ± SD. **c** BMDMs were pretreated with $PGE_2$ for 30 min and then treated with LPS for 4 h. Lysates were assessed with caspase-11 expression by western blotting. A representative western blot from three independent experiments is shown. **d** BMDMs from wild-type or EP2-deficient mice were assessed for caspase-11 expression by western blotting. The results shown are representative of BMDMs prepared from three mice of each genotype in a single experiment. Densitometry values are shown above each blot.

cell numbers, specifically infiltrated T cells, eosinophils, and neutrophils (Fig. 4a–d, gating strategy provided in Supplementary Fig. 1). Misoprostol-treated mice had decreased caspase-11 and IL-1β levels in the lung (Fig. 4f, compare lanes 5–8 to 9–12, quantified in Fig. 4g). Furthermore, the NSAID indomethacin, which blocks among other prostanoids $PGE_2$ production, significantly enhanced expression of caspase-11 (Fig. 4h, compare AAI with AAI+Indo groups, quantified in Fig. 4i) in the lungs of asthmatic mice. This result suggests that enhanced caspase-11 expression is also regulated by endogenous $PGE_2$ production.

**Caspase-11-deficient mice are protected in a model of allergic lung inflammation.** To further assess the role and importance of caspase-11 in asthma, caspase-11-deficient mice were tested in the AAI model. Caspase-11$^{-/-}$ mice had decreased levels of leukocyte numbers in bronchoalveolar lavage fluid, and differential staining of lung lavage demonstrated that caspase-11$^{-/-}$ mice had fewer infiltrating alveolar eosinophils (Fig. 5a). Moreover, another hallmark of AAI, namely an increase in Th2 cytokines, was either completely abolished (IL-4) or significantly diminished (IL-5) (Fig. 5b). H&E staining of lung sections demonstrated decreased cell infiltration in the area of large bronchi (Fig. 5c). Flow cytometry analysis (gating strategy provided in Supplementary Fig. 2) of the lung tissue revealed decreased percentage and total cell number of lung eosinophils (Fig. 5d), as well as T cells (Fig. 5e). Intracellular staining of infiltrated T cells demonstrated that caspase-11$^{-/-}$ mice had also impaired production of Th1 and Th17 cytokines (Fig. 5f). Analysis of serum isolated from mice after the second airway

challenge has also shown a significant reduction in circulating IgE levels (Fig. 5g). Altogether, these results clearly show that caspase-11-deficient mice are protected in AAI, and an adaptive immune response is not mounted, suggesting that caspase-11 is required for key features of allergic airway inflammation. Finally, we also found that caspase-4 was upregulated in alveolar macrophages isolated from asthma patients compared with healthy controls (Fig. 5h).

## Discussion

In this study, we report the endogenous inhibitor of caspase-11. $PGE_2$ blocks the induction of caspase-11 by LPS by inhibition of IFN-β production. IFN-β production is necessary for caspase-11 induction by LPS[26]. We have also found that $PGE_2$ inhibits caspase-11-driven pyroptosis following induction of caspase-11 expression by LPS. $PGE_2$ contributes to the resolution phase of inflammation, and has been known for years to have a protective role in tissue homeostasis[10,31]. Our data demonstrate that inhibition of caspase-11-mediated pyroptosis may be important for these tissue-protective effects. Although we speculate here that the protective effect of $PGE_2$ in asthma can be explained by the ability of $PGE_2$ to inhibit pyroptosis and caspase-11/4 expression, we have not provided in vivo evidence for this axis to occur. $PGE_2$ can be protective in our in vivo model in multiple ways, including inhibition of eosinophil trafficking[30], vascular remodeling[32], group 2 innate lymphoid cells[33], mast cells[18,34], and T cells[20].

Asthma is a chronic inflammatory disease of the airways characterized by reversible airflow obstruction and inflammatory cell infiltration, including monocytes, which contribute to disease

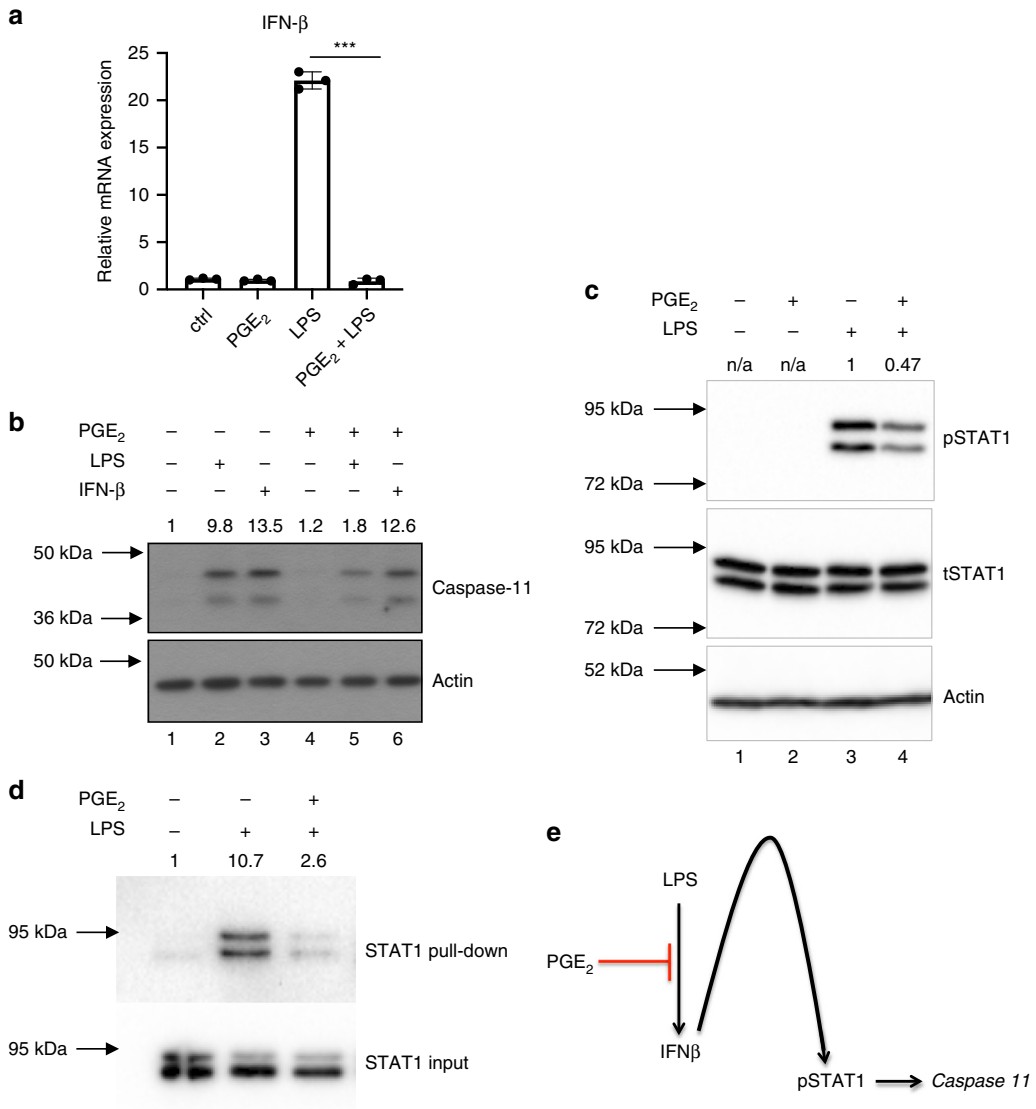

**Fig. 2 Prostaglandin E₂ inhibits caspase-11 expression by inhibition of IFN-β production. a** BMDMs were treated with 1 μM PGE₂ or DMSO for 30 min followed by 100 ng/ml LPS for 4 h, and subjected to qPCR analysis for IFN-β. Data from BMDMs from three mice are shown, and are representative of three independent experiments, each involving three mice, with individual data points on the graph, which are means from technical duplicates. ***$P < 0.0001$, two-tailed Student's *t* test, error bars represent mean ± SD. **b** BMDMs were treated with 1 μM PGE₂ for 30 min followed by 100 ng/ml of LPS or IFN-β for 4 h, and assessed for caspase-11 by western blotting. The results shown are representative of three independent experiments. **c** BMDMs were treated with 1 μM PGE₂ for 30 min followed by 100 ng/ml of LPS for 2 h, and assessed for phosphor-STAT-1 by western blotting. The results shown are representative of three independent experiments. **d** BMDMs were treated with 1 μM PGE₂ or DMSO for 30 min followed by 100 ng/ml of LPS for 2 h. An oligonucleotide pulldown using the caspase-11 promoter was then carried out with samples and then immunoblotted for STAT-1. The results shown are from a single experiment. Densitometry values are shown above each blot. **e** Schematic representation of PGE₂-mediated inhibition of caspase-11 transcription.

severity[35]. Like most complex diseases, asthma is believed to be caused by a combination of genetic and environmental factors. Exposure to a wide range of microbes, in line with the hygiene hypothesis, inversely correlates with asthma[36]. However, known inducers of caspase-11, such as bacterial infections and endotoxin, were demonstrated to exacerbate asthma in humans[37,38], and LPS was shown to worsen[39,40] or even be necessary[41] to induce asthma in animal models. We do not know how caspase-11 is being induced in the allergic airway inflammation model. It is possible, however, that the protective nature of PGE₂ in the model of allergic airway inflammation is caused by the inhibition of LPS-induced caspase-11 expression.

Another common environmental factor involved in asthma exacerbations is the use of NSAIDs, which inhibit prostanoid production, and are well known to exacerbate asthma in over 10% of asthmatics[14,42]. Furthermore, genetic deletion or pharmacological inhibition of COX enzymes by indomethacin increases allergic lung inflammation[43,44] in murine models of asthma. By showing that endogenously produced PGE₂ is inhibitory for caspase-11 expression, we provide a putative explanation for the NSAID-driven exacerbation in asthma[15]. Caspase-4/11 expression is therefore altered beyond bacterial infections, in this case during treatment with NSAIDs. Obtaining caspase-4 and IL-1β expression data from trials examining the effects of NSAIDs, such as a recent low-dose Aspirin trial[45], might confirm the physiological relevance of our findings. Although there is extensive literature, including in human studies[14], which describe the protective nature of PGE₂ in asthma[46], there is also evidence for

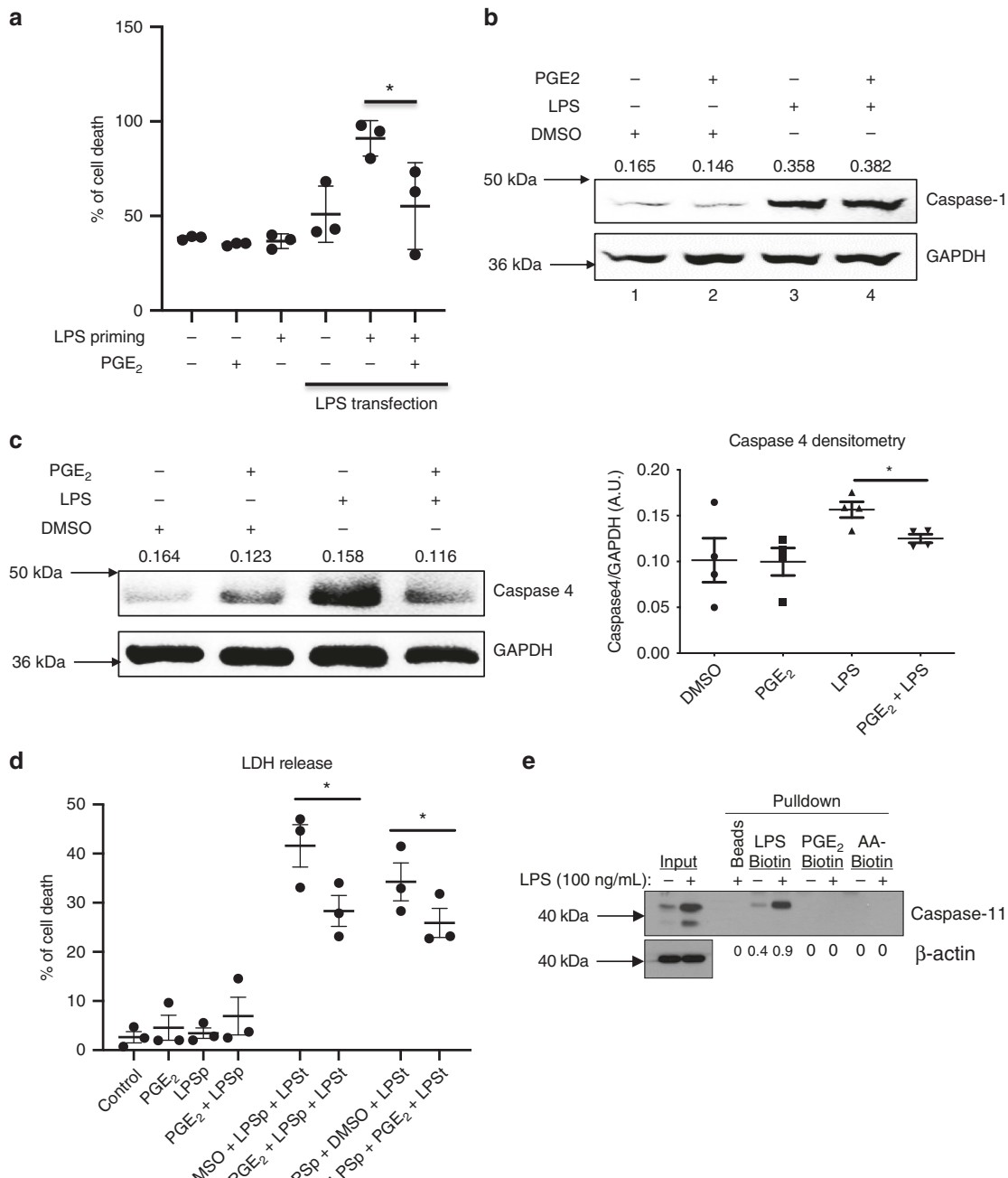

**Fig. 3 Prostaglandin E2 inhibits capase-11-driven pyroptosis in LPS-primed cells. a** Murine BMDMs were primed with 100 ng/ml of LPS for 4 h followed by stimulation with 1 μM PGE2 for 30 min; media was removed and followed by 2 μg of LPS transfected using FuGENE 0.25% v/v liposomes overnight. Cell death was assessed by measuring LDH activity. The results shown are from a single experiment, with three mice in each group, and are representative of three independent experiments. Individual data points on the graph are means from technical duplicates. Ordinary one-way ANOVA with Tukey's multiple comparison test has been used, *$P < 0.01$, error bars represent mean ± SD. **b** BMDMs were primed with 100 ng/ml of LPS for 4 h followed by stimulation with 1 μM PGE2 for 30 min, and lysates were assessed for caspase-11 expression by western blotting. The results shown are representative of three separate experiments. **c** Human monocyte-derived macrophages were treated with 1 μM PGE2 or DMSO for 30 min followed by 100 ng/ml LPS for 4 h. Lysates were assessed for caspase-4 expression by western blotting. The results shown are representative of three independent experiments, each utilizing one or two donors. Densitometry from all four donors is provided (*$P < 0.01$, two-tailed paired Student's $t$ test, error bars represent mean ± SEM). **d** Human monocyte-derived macrophages were treated with 1 μM PGE2 for 30 min before priming with 100 ng/ml of LPS for 4 h or for 30 min before transfection with 2 μg/mL of LPS using FuGENE 0.25% v/v liposomes o/n. Supernatants were collected and analyzed for cell death using LDH assay. Data from three independent experiments are shown, each experiment utilizing cells from one donor. Individual data points on the graph are means from technical duplicates (*$P < 0.01$, two-tailed paired Student's $t$ test, error bars represent mean ± SEM). **e** Streptavidin pull-down assay. BMDMs were treated with 100 ng/ml of LPS for 4 h, lysed, precleared with strepatvidin beads and incubated with 2 μg of biotinylated LPS, PGE2 and arachidonic acid, and streptavidin beads for 1 h, and blotted for caspase-11. The results shown are representative of two separate experiments.

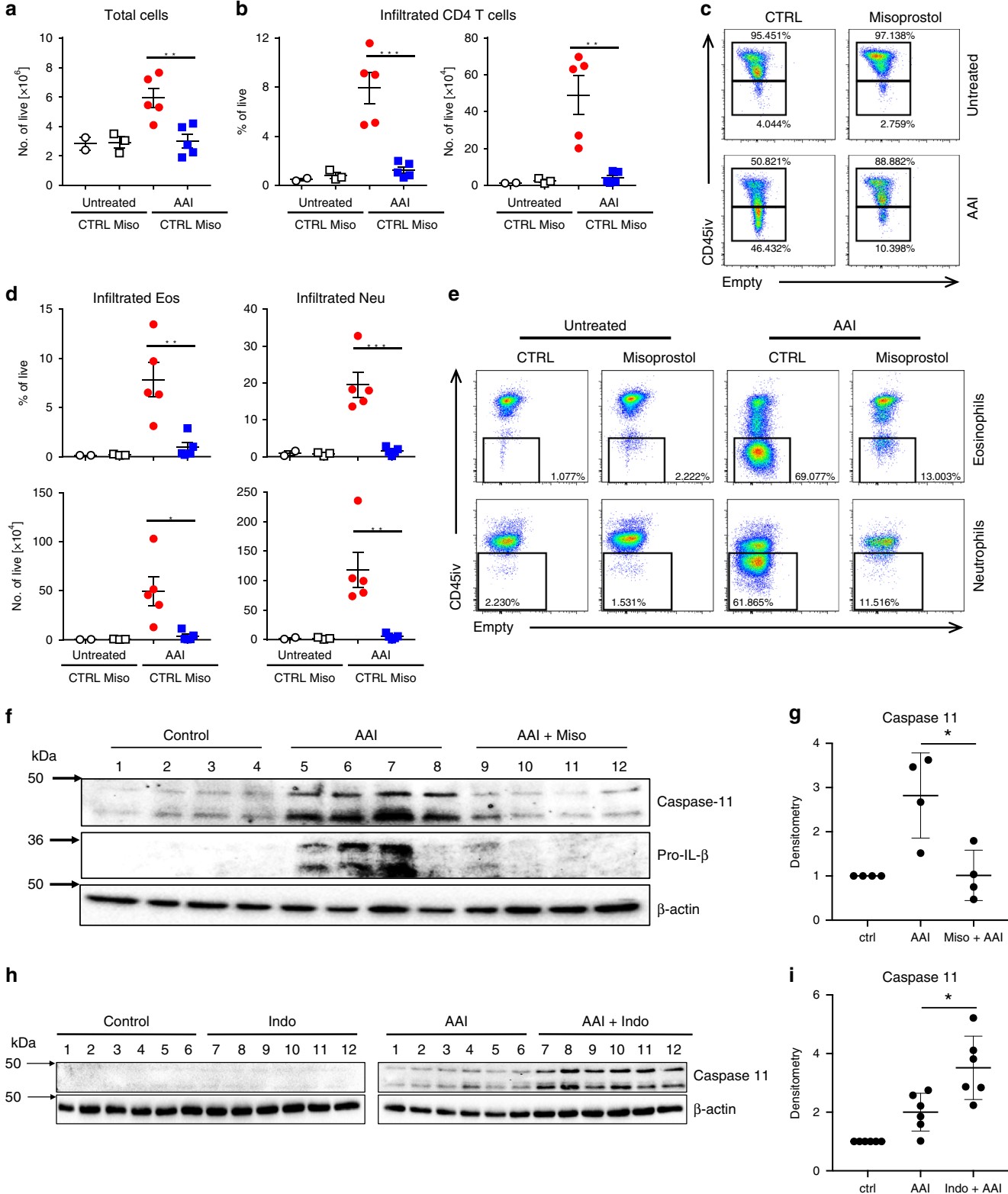

PGE$_2$–EP2 signaling contributing to asthma pathogenesis, by promoting IgE production[47]. These controversial findings highlight the ability of PGE$_2$ to exert opposite effects on different cell types, which can often lead to opposing results in animal models.

Although we use LPS to induce caspase-11/4 in vitro, we can only speculate that in asthma, LPS is a driver of caspase-11/4 expression. Lung is no longer considered a sterile environment, and harbors a variety of bacteria including LPS-containing microbes[48]. Exposure to LPS has also been shown to play a major role in asthma development[49]. LPS in our model might come from naturally occurring lung bacteria, and can activate caspase-11 upon cell damage caused by inflammation. There is also a possibility of LPS contaminating the ovalbumin used. Asthma and COPD symptoms are often exacerbated by infections, and damage to the lung cells leads to persistent inflammation airway remodeling[50] and activation of NLRP3[51]. The lung is an organ

**Fig. 4 Indomethacin induces whereas misoprostol decreases caspase-11 expression in allergic airway inflammation. a–e** Allergic airway inflammation (AAI) was induced by intraperitoneal injection of 20 μg of OVA mixed with 2 mg of alum, followed 7 days later by two airway challenges with 1% OVA and lung lysate collection 24 h after the final airway challenge. In all, 2 mg/kg of misoprostol was given 2 h before sensitization and 2 h before each airway challenge. Lungs were homogenized, and cellular infiltration as well as cytokine production was analyzed using flow cytometry. To discriminate blood-borne circulating cells from lung-localized cells, we used CD45 i.v. administration 10 min before the mouse was killed and lungs were harvested. Data are from one experiment using five mice per group (*P < 0.01, **P < 0.01, ***P < 0.001, two-tailed paired Student's t test, error bars represent mean ± SD). **f** The total lung lysates, obtained by homogenization of whole-lung lobes with tissue grinder, were probed for caspase-11 and IL-1β expression. There were four mice in each group, and analysis of samples from each mouse is shown. **g** Densitometry from the expression of caspase-11 presented in panel F (*P < 0.01, two-tailed paired Student's t test, error bars represent mean ± SD). **h** In total, 2 mg/kg of indomethacin was given 2 h before sensitization and 2 h before each airway challenge. Total lungs lysates were probed for IL-1β and caspase-11 expression, with six mice in each group. Analysis of samples from each mouse are shown. Data are from two independent experiments, each with three mice per group. **i** Densitometry from the expression of caspase-11 presented in panel **h** is shown (*P < 0.01, two-tailed paired Student's t test, error bars represent mean ± SD).

where regulation of an inflammatory cell death process, such as pyroptosis, is extremely important because of its delicate structure. While cell death in asthma has been quite intensively studied, researchers focused predominantly on the role of apoptosis[52]. It has been recently shown that common allergens can drive pyroptosis and contribute to asthma[53]. Taken with the effect of NSAIDs presented here, and previous data demonstrating a beneficial effect for a global caspase inhibitor[54], limiting pyroptosis becomes even more relevant as a therapeutic strategy for asthma and possibly other respiratory diseases, such as COPD. However, since asthma is related to airway fungal infections[55], and allergic inflammation is considered an elaborate antifungal program[56,57], inhibition of caspase-4/11 might however compromise antifungal immunity and possibly worsen disease, given that yeasts and molds can both activate caspase-11 and trigger pyroptosis[58,59].

Inhibition of caspase-11-mediated pyroptosis by $PGE_2$ will decrease the release of DAMPs such as ATP, DNA, or HMGB1. Although caspase-11-driven pyroptosis is NLRP3 independent[60], we cannot exclude the possibility that independently of pyroptosis, $PGE_2$ exerts an additional effect on NLRP3, by decreasing the release of DAMPs, which might significantly blunt activation of NLRP3 by DAMPs, and might therefore explain why caspase-11 deficiency or its inhibition by $PGE_2$ might be so efficient in vivo in blocking inflammation. Both autophagy and apoptosis are preferable to inflammatory cell death for successful resolution of inflammation[61,62]. It is therefore reasonable to speculate that $PGE_2$ not only inhibits pyroptosis, but also induces apoptosis, as was demonstrated in other models[63–66]. Determining the role of endogenous $PGE_2$ in limiting caspase-11-mediated cell death could be further tested in EP2-deficient mice. Importantly, $PGE_2$ was shown previously to inhibit necrosis and promote apoptosis in *Mycobacterium tuberculosis*-infected alveolar macrophages, which proved to be beneficial to the host[67,68]. Macrophages from caspase-1/11 double-knockout mice are protected from pyroptosis and instead undergo autophagy upon infection[69,70], and we believe that $PGE_2$ might promote similar cellular reprogramming.

The protective nature of $PGE_2$ on tissue homeostasis might therefore be in part explained by the ability of $PGE_2$ to block the expression of caspase-4/11. This may have particular relevance to asthma. Figure 6 illustrates the main findings of our work. Our findings therefore define a role for caspase-11 in allergic airway inflammation, and support inhibition of caspase-4 as a therapeutic strategy in asthma.

## Methods
**Mice.** Casp11$^{-/-}$ mice on the C57BL/6J background were obtained from J. Yuan's laboratory (Harvard Medical School, USA), and were subsequently backcrossed onto the C57BL/6J (Harlan Laboratories, UK) background for another eight generations. Heterozygous breeding pairs were used to generate wild-type (WT) and Casp11$^{-/-}$ littermates, which were used for all experiments described. Experiments were performed with 8-to-12-week-old female mice bred under specific pathogen-free conditions, under license and approval of the local animal research ethics committee (Health Products Regulatory Authority).

PCR was used to genotype caspase-11-knockout mice. Genomic DNA was isolated from tails. The primer sequences used for the PCR were as follows: SY-21, 5′-GGCATGGAGTCAGAGATGAAAGAC-3′; SY-22, 5′-GCCCATGTGGCATTACCTGCCAGC-3′; SYKO, 5′-AGATCTACACCTCTGCACAACTGGGGT-3′; PJK, 5′-TGGCGCTACCGGTGGATGTGGAATGTG-3′. The wild-type genome of caspase-11 could be detected using SY-21 and SY-22 (an ∼200-bp PCR product), and the mutant caspase-11 gene could be detected using SYKO and PJK (an ∼600-bp PCR product)[71].

**Cell culture.** Bone marrow-derived macrophages (BMDMs) were isolated and cultured in 20% L929 media until day 6, after which they were plated for experimentation. All experiments were carried out with prior ethical approval from the Trinity College Dublin Animal Research Ethics Committee. PBMCs from healthy volunteers were isolated from buffy coats obtained from the blood transfusion services in St. James's Hospital (Dublin, Ireland). PBMCs were isolated by Ficoll/metrizoate density gradient centrifugation (Lymphoprep; Nycomed, Marlow, UK). Monocytes were obtained by plastic adherence and cultured in 10% human serum for 5 days to obtain monocyte-derived macrophages.

**Reagents.** RPMI 1640 culture medium and penicillin/streptomycin/amphotericin B solution were purchased from Invitrogen (Carlsbad, CA). $PGE_2$ and misoprostol were purchased from Cayman Chemical (Ann Arbor, MI); DMSO served as vehicle control. The protein kinase A (PKA)-specific cAMP analog 6-Bnz-cAMP was purchased from Biolog (Bremen, Germany). LPS was purchased from Sigma-Aldrich (St. Louis, MO).

**RNA isolation and quantitative real-time PCR.** RNA extraction was performed using the RNeasy mini kit (Qiagen), and cDNA was generated using an Applied Biosystem high-capacity cDNA archive kit. The quantitative real-time PCR analysis was performed with an ABI 7500 Fast real-time PCR system (Applied Biosystem). Reactions were set up with SYBR Green PCR core reagents (Invitrogen). Data were normalized to beta-actin, and mRNA expression fold change relative to controls was calculated using the $2^{-\Delta\Delta Ct}$ method. The following primers were used: Cas 11F: 5′-CCT GAA GAG TTC ACA AGG CTT-3′ and R: 5′-CCT TTC GTG TAG GGC CAT TG-3′; IFN-β F: 5′-ATG TGT GTC CGA GCA GAG AT-3′ and R: 5′-CCA CCA CTC ATT CTG AGG CA-3′; Cas 4F: 5′-GCT CTT CAA CGC CAC ACA AC-3′ and R: 5′-GGT GGG CAT TTG AGC TTT GG-3′; Cas 5F: 5′-AGG CCT GCA GAG GTG AAA AA-3′ and R: 5′-TGA AGA TGG AGC CCC TTG TG-3′; hGSDMD F: 5′-CTG CTA GAA CCC AGG ATC GC-3′ and R: 5′-CAT GCT CCG TGA CCG TCG-3′; beta-actin, F: 5′-ACC CTA AGG CCA ACC GTG A-3′ and R: 5′-CAG AGG CATA CAG CGA CAG CA-3′; hPBGD forward: F: 5′-ACC CTA GAA ACC CTG CCA GAG AA-3′ and R: 5′-GCC GGG TGT TGA GGT TTC CCC-3′.

**STAT-1 binding to caspase-11 promoter.** Biotinylated oligonucleotides containing the STAT-1 binding site on the caspase-11 promoter were annealed at 90–95 °C for 3–5 min and allowed to cool to room temperature (Forward BIO: 5′-CTT TCA ACA TCT CCT GGA AGT CCC CG-3′, Reverse: 5′-CGG GGA CTT CCA GGA GAT GTT GAA AG-3′)[26]. After stimulation with LPS (+/− $PGE_2$), BMDMs were lysed in 100 μl of oligonucleotide buffer (25 mM Tris, 5% glycerol, 50 mM EDTA, 5 mM NaF, Nonidet P-40 1%, 1 mM DTT, 150 mM NaCl, and protease and phosphatase inhibitors), and snap-frozen. Samples were then thawed on ice and diluted with a further 900 μl of oligonucleotide buffer containing no NaCl. In total, a 10-μl sample of lysate was kept to which 40 μl of 5× SDS buffer was added. The remaining lysates were then precleared with 20 μl of prewashed streptavidin–agarose beads, rotating at 4 °C for 15 min before centrifuging at 2500 rpm for 5 min at 4 °C. Supernatants were removed in a fresh tube with 30 μl of

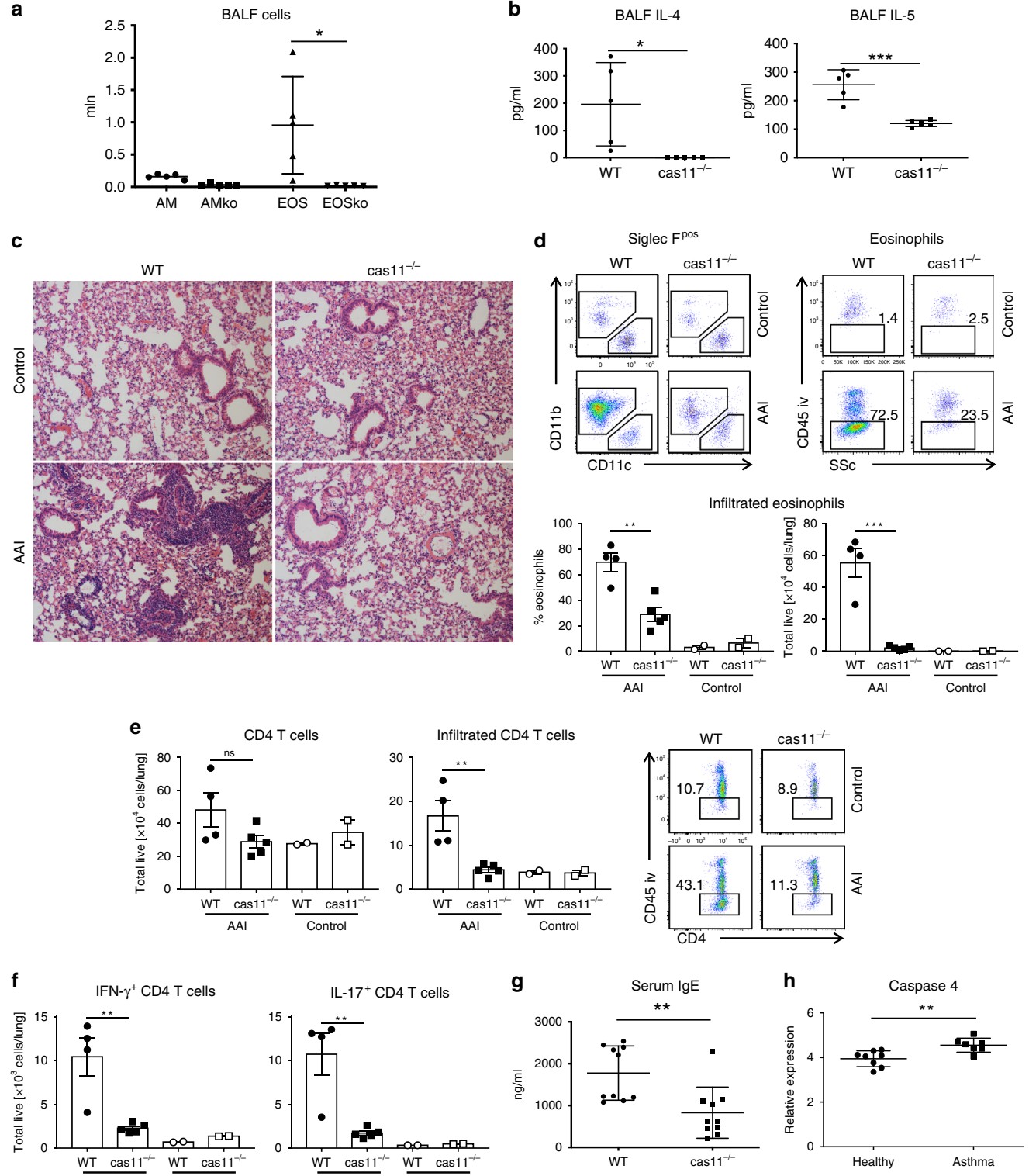

prewashed streptavidin–agarose beads and 30 μg of 5′-biotinylated oligonucleotide. Binding was performed for 2 h at 4 °C, by rotating. Samples were centrifuged for 5 min at 4 °C to pellet the beads, which were washed three times before 50 μl of 5× SDS sample buffer was added to the beads[72]. STAT-1 binding to caspase-11 promoter was detected by western blotting, and bands were visualized using Gel Doc™ EZ System gel imaging system.

**Cell death assay**. Cell death was measured with The CytoTox-ONE™ Assay (Promega), which is a fluorescent measure of the release of lactate dehydrogenase (LDH) from cells, with a damaged membrane into the culture medium. Cell death was presented as percentage, and lysed cells served as 100%, and medium as 0%.

Values of the percent of cell death, measured by LDH release to the supernatant, presented in the paper, are relative and differ between experiments.

**Induction of caspase-11-dependent pyroptosis**. LPS was transfected overnight at a concentration of 2 μg/ml after 4 h of priming with 100 ng/ml of LPS using liposome 0.25% v/v FuGENE HD (Promega)[60].

**Human data**. Patients with clinically confirmed bronchial asthma and healthy volunteers were recruited by the Clinic for Internal Medicine—Department for Pneumology, University Medical Center Marburg. Human subjects participating in

**Fig. 5 Caspase-11 drives allergic airway inflammation.** WT (white bars) and caspase-11-deficient (black bars) mice were subjected to allergic airway inflammation protocol as explained below. Briefly, allergic airway inflammation was induced by intraperitoneal injection of 20 µg of OVA mixed with 2 mg of alum, followed 7 days later by two airway challenges with 1% OVA and lung lysate collection 24 h after the final airway challenge. To discriminate blood-borne circulating cells from lung-localized cells, we used CD45 i.v. administration 10 min before the mouse was killed and lungs were harvested. Lung lavage fluid was collected and subjected to (**a**) differential cell count. **b** ELISA analysis. **c** H&E staining of lung sections. Lungs were homogenized, and cellular infiltration as well as cytokine production was analyzed using flow cytometry (**d–f**). Data for each mouse tested are shown, with 4–5 mice per group, and are represenative of three independent experiments. *$P < 0.05$, **$P < 0.01$, ***$P < 0.001$ (two-tailed Student's $t$ test, error bars represent mean ± SD). **g** Blood was collected from control and OVA-treated mice 24 h after the final airway challenge, and serum was isolated and analyzed for IgE levels by ELISA. Two independent experiments were carried out with five mice per group; individual data points on the graph are means from technical duplicates, **$P < 0.01$ (two-tailed Student's $t$ test, error bars represent mean ± SD). **h** Alveolar macrophages from asthmatic or control patients were subjected to qPCR analysis for caspase-4 (eight healthy and seven asthmatic patients, data points on the graph are means from technical duplicates, **$P < 0.01$, Mann–Whitney test. Error bars represent mean ± SD).

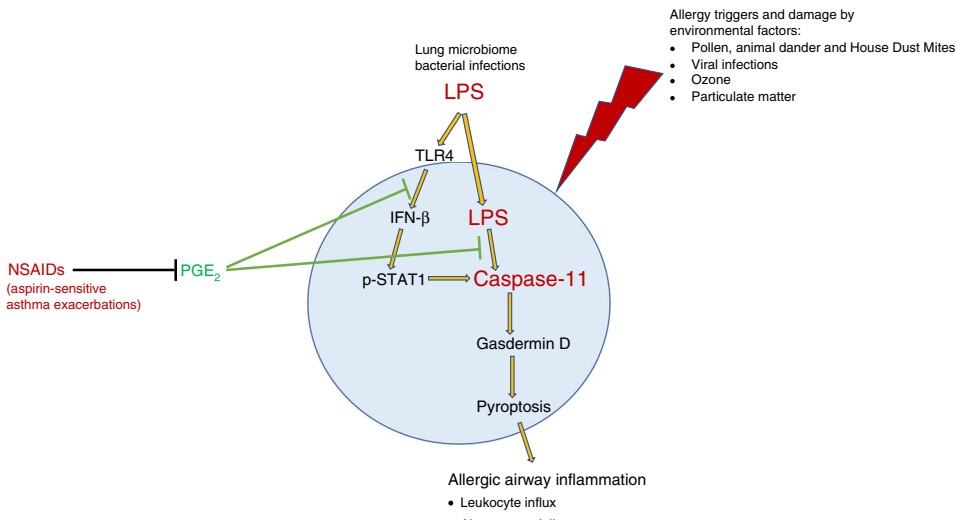

**Fig. 6 Schematic representation of the protective role of PGE2 in inhibition of caspase-11 and allergic airway inflammation.** In the lung, extracellular LPS from the lung microbiome or Gram-negative bacterial infections triggers LPS signaling, which induces IFN-β production. IFN-β in turn activates STAT-1, which gets phosphorylated and initiates transcription of caspase-11. Intracellular LPS activates caspase-11, which initiates a highly inflammatory process of cell death termed pyroptosis. Pyroptosis is therefore likely to participate or even initiate allergic airway inflammation. PGE$_2$ can inhibit IFN-β production and caspase-11 transcription, as well as caspase-11 activation by LPS, thereby limiting pyroptosis.

this study signed informed consent to participate in this research. Asthma was diagnosed by symptoms in conjunction with airway hyperresponsiveness (metacholine challenge). Furthermore, the concentration of nitrogen monoxide (FeNO) for all patients and controls was measured. BALF was obtained following the ATS consensus procedure in accordance with local ethics regulations (87/12) from healthy subjects and asthma patients. The study was approved and oversaw by Ethics Committee of the Medical Faculty of the Philipps University, Marburg, Germany. Alveolar macrophages (AM) were isolated from the BAL fluid by negative selection as described for sputum macrophage[73]. RNA was isolated from the AM fraction by phenol/chloroform extraction.

**Western blot**. Cells were lysed in 5× sample buffer and separated by SDS-PAGE and blotted according to standard protocols. For measurement of cleaved IL-1β and caspase-1 in the supernatant, proteins in the supernatant were precipitated with 1% (v/v) StrataClean resin (Agilent Technologies), and beads were lysed in 5× sample buffer. Proteins were visualized using the HRP substrate WesternBright ECL spray (Advansta) on a ChemiDoc imaging system (Bio-Rad), apart from blots presented in Figs. 2b and 3e, which were developed using traditional X-ray film. All uncropped and unprocessed scans of the most important blots are available in the Source Data file. Primary Abs were β-actin (1:15,000, AC-74, Sigma-Aldrich), Gapdh (1:5000, 6C5, Calbiochem), IL-1β (pro- and cleaved, 1:1000, AF-401, R&D Systems), caspase-11 (1:1000, 17D9, Sigma-Aldrich), p-STAT-1 (1:1000, #9177, Ser727, Cell Signalling) and total STAT-1 (1:1000, #9172, Cell Signalling), and caspase-4 (4B9, MBL). HRP-conjugated secondary Abs were from Jackson ImmunoResearch Laboratories.

**ELISA**. IL-4, IL-5, IgE, and IL-1β concentrations in supernatants were measured by ELISA according to the manufacturer's instructions (R&D Systems).

**Histology**. Lungs were rapidly removed from mice and dipped in 10% neutral buffered formalin (Diapath, Italy) for 24 h. Lung tissues were transferred to 70% ethanol following paraffin wax embedding that was performed using an automatic tissue processor (Leica Microsystems). Five-micrometer-thick lung sections were stained with hematoxylin and eosin.

**OVA-induced allergic airway inflammation**. Allergic airway inflammation was induced by means of intraperitoneal injection of 20 µg of OVA (Sigma) mixed with 2 mg of alum (Thermo Scientific, Waltham, MA), followed 7 days later by two airway challenges with 1% OVA. This well-established protocol is known to result in eosinophilic inflammation and induction of TH2 cytokines in bronchoalveolar lavage fluid (BALF). Samples were collected 24 h after the last airway challenge. Total cells were counted, followed by differential counting of Wright–Giemsa-stained cytospin preparations. Lavage fluid recovered from the first 0.6-mL aliquot injected into the lung was analyzed by means of ELISA to assess cytokines.

**In vivo misoprostol and indomethacin treatment**. Misoprostol was administered in vivo according to an established protocol[20]. Mice were injected subcutaneously with 200 µL of saline containing 2 mg/kg of indomethacin or misoprostol in 0.5% DMSO 2 h before intraperitoneal sensitization and airway challenge with OVA; control mice received 200 µL of saline containing 0.5% DMSO alone.

**Isolation and FACS analysis of cells from control and asthmatic lung**. Lung tissue was chopped and digested with collagenase D (1 mg/ml, Roche) and DNase I (10 µg/ml, Sigma-Aldrich) for 1 h at 37 °C with agitation. Next, lungs or spleens were passed through a 40-µm cell strainer to obtain a single-cell suspension, followed by RBC lysis. The cells were incubated with CD16/CD32 FcγRIII (1:100) to block IgG Fc receptors. Cells were incubated with LIVE/DEAD Aqua (Invitrogen), followed by surface staining with fluorochrome-conjugated anti-mouse Abs for various markers. To detect cytokines, cells were stimulated with PMA (50 ng/ml)

and ionomycin (500 ng/ml) in the presence of brefeldin A (5 μg/ml) for 4 h at 37 °C. For detection of intracellular cytokines, cells were fixed in 2% PFA and permeabilized with 0.5% saponin (Sigma-Aldrich, Ireland), followed by staining with IL-17A–V450 and IFN-γ–PE-CF594 (BD Biosciences). To discriminate blood-borne circulating cells from lung-localized cells, we used a well-described approach in which anti-mouse PE-CD45 Ab (eBioscience) was administered i.v. to mice 10 min before they were euthanized and lungs were harvested[74]. Circulating leukocytes, which are exposed to the antibody and are labeled, become CD45iv⁺, whereas tissue-infiltrated cells are "protected" from labeling and remain CD45iv⁻. Gating strategy is provided in the Supplementary Information, as well as Source Data files.

**Statistical analysis and data presentation**. Statistical significance was analyzed using the GraphPad Prism 5.0 statistical program (GraphPad Software, La Jolla, CA). Graphs show individual data points calculated from technical replicates; means are depicted and error bars represent SEM.

**Reporting summary**. Further information on research design is available in the Nature Research Reporting Summary linked to this article.

## Data availability

The data that support the findings of this study are available from the corresponding author upon reasonable request. Western blots and graphs shown in the main figures are available in the Source Data file.

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

## Acknowledgements

We would like to thank Dr. Michael Carty for all the help with his expertise in the inflammasome assay.

## Author contributions

Z.Z. designed and conducted experiments, analyzed and interpreted data, and wrote the paper; E.F., R.G.C., E.M.P.D., M.M.H., C.D., K.B., D.G.R., A.H., A.M., J.K., and G.L. conducted in vitro experiments and analyzed data; M.W. helped with in vivo experiments and conducted flow cytometry experiments and analysis; W.B., T.G., and B.S. provided human samples; O.J.M., K.H.G.M., M.W., E.C.L., and E.M. helped to design experiments and provided critical input; L.A.J.O. conceived the study, funded and oversaw the research program, and wrote the paper.

## Competing interests

The authors reviewed and approved the final version of the paper. The authors declare no competing interests.
