## [Peer Review File · Nature Communications]

Reviewers' comments:

Reviewer #1 (Inflammasome, cell death)(Remarks to the Author):

Zaslona et al. explore the role of the anti-inflammatory prostaglandin PGE2 in regulating the inflammatory caspase-11 and the role of caspase-11 in asthma. Using in vitro assays, they demonstrate the PGE2 decreases the ability of LPS priming to induce caspase-11 expression likely by downregulating TLR4 driven IFN β production. Additionally, they argue that PGE2 causes phosphorylation of caspase-11 at serine-350, shown by mass spectrometry. They argue that this is a second way that PGE2 may inhibit caspase-11, explored using transfection of mutant caspase-11. Furthermore, they observed that their caspase-11 $-/-$ mice had decreased levels of various immunologic parameters in the lungs of mice in an asthma model.

Ultimately, the largest weakness of this paper is a lack of significant impact/novelty, which may be partly due to the fact that this paper reads as two small papers strung together into a single paper whereby the first story (regulation of caspase-11 by PGE2) seems only peripherally connected to the second story (caspase-11 in asthma) by the observation that PGE2 helps in asthma. The conclusion that PGE2 inhibits caspase-11 expression by inhibiting autocrine IFN β is expected from published literature (PGE2 is known to inhibit IFN β expression and IFN β expression is known to drive caspase-11 expression already). Conversely, the significance of phosphorylation of caspase-11 needs to be more firmly demonstrated. Finally, while clearly establishing a role for caspase-11 in asthma would be significant, more of this particular story would need to be established (eg how is caspase-11 activated) and issues of potential confounders (lung microbiota, potential LPS contamination, etc) would need to be more clearly addressed to make the asthma data publishable. In summary, this manuscript is not suited for publication at all, and an immense amount of work remains to be done.

Major points:

There seems to be no hypothesis that forms the premise of this paper. Why did the authors think that PGE2 would inhibit caspase-11 in the first place?

We have known at least since 2008 that PGE2 inhibits TLR4 driven type I IFN production. Since caspase-11 is dependent upon this, it is not novel to show the link again. However, this part of the paper could be published as a PLOS One paper, verifying expected results.

Prostaglandin E2 Suppresses Lipopolysaccharide-Stimulated IFN- β Production

X. Julia Xu, Jonathan S. Reichner, Balduino Mastrofrancesco, William L. Henry, Jr. and Jorge E. Albina
J Immunol 2008; 180:2125-2131

doi: 10.4049/jimmunol.180.4.2125

Fig. 3, the cells are still being exposed overnight to TLR4 stimulating LPS in the context of overnight PGE2, so this really cannot rule out that TLR4 signaling is being affected by the PGE2 continuously overnight.

Fig. 3 shows that PGE2 causes detectable caspase-11 phosphorylation, however this may not have any physiologic relevance. It may be only 1% of the available caspase-11 protein pool that is phosphorylated, which often happens with kinases, and this may no real importance.

Regarding the 293 overexpressed point mutants in Fig. 3, while their data suggest a potential role of phosphorylation, they do not clearly demonstrate that phosphorylation as opposed to some other issue (eg difference in folding affecting the auto-activation properties of the protein when the protein is overexpressed) as the cause of the observed mild difference in LDH release. There is no treatment with PGE2 here, and there is no activation of caspase-11 by LPS. Thus these experiments are quite non-physiologic.

In figure 3A, why is % cell death of cells so much less than in figure 1A LPS-transfected, PGE2 untreated controls (~20% vs ~80%) despite what appears to be only minor variation in protocol?

While the authors demonstrate a difference in immunologic parameters within the lung in Fig. 4 and one would expect that these parameters correlate with asthma phenotype, the authors should more directly demonstrate an effect on airway hypersensitivity (eg methacholine challenge). One wants to actually show that the gene affects the clinical phenotype.

The idea that endogenous lipids activates caspase-11 (reference 5) is extremely controversial, and cannot be taken as a given for this study. Thus, it remains unclear how caspase-11 would be activated in the asthma model, where is the LPS?

There have been conflicting evidence regarding the roles of inflammasomes and downstream IL-1b in mouse models of asthma. In light of this fact and the fact that the mechanism by which caspase-11 is activated in this model is not addressed, the authors need to address potential confounders such as contributory effects of differing lung microbiota. They need to use littermate controlled mice – this is essential, very important.

They also need to consider the possibility of LPS contamination of reagents.

Figure 5 is not definitive that the PGE2 effect on caspase-11 is occurring in the asthma model. It is only correlative. This drives home that there is really a disconnect between the two stories: the one that PGE2 affects caspase-11, and the other that caspase-11 is involved in asthma.

Minor points.

Introduction stating that PGE2 is the primary target of NSAIDs. I don't think that is correct, since NSAIDs inhibit COX.

Fig. 2 is explained incorrectly. the STAT1 phosphorylation is downstream of IFN- γ . Thus, the inhibition of IFN- γ by PGE2 explains the lower phospho STAT1 in Fig. 2C-D, which is not the impression given by the text.

Reviewer #2 (Airway inflammation, Th function)(Remarks to the Author):

The investigators have evaluated the role of caspase -11 in the expression of allergic airway disease and the inhibitory effect of PGE2. The data are excellent, as far as they go, and the message is very interesting and important, but there are significant issues to overcome at the same time:

1. Western blots: densitometric quantification is helpfully employed for some westerns. Please expand use to all such blots to enhance understanding of the semiquantitative changes.
2. Figure 5. It is confusing to label conditions as "asthma" when this term is only properly used in the context of human disease. Suggest referring to mouse disease as "allergic airway disease" (AAD) or something similar to avoid confusion.
3. Figure 5A: there are doublets for IL-1b and cas11-please add arrows to indicate what are the true bands of interest.
4. Please describe what a "lung lysate" is and how they are prepared.
5. The authors exaggerate by saying that in the absence of caspase 11 an adaptive immune response is not mounted-it is reduced, but not abrogated.
6. Allergic airway disease model: this is not an asthma model as there are no measurements of airway mechanics, e.g., airway responsiveness to methacholine or acetylcholine challenge. Please add these

data if these measurements are possible. If the data cannot be obtained, the model should be referred to as something else, such as "model of eosinophilic lung inflammation, etc.

7. The ovalbumin model has faded in relevance in recent years with the gradual understanding that asthma and related diseases are proteinase-activated diseases. More confidence in the in vivo data would be possible by repeating experiments in which the allergic airway disease was induced by a proteinase allergen (e.g., proteinase of *Aspergillus oryzae* or *A. melleus*), or a whole fungal, whole dust mite, or whole pollen extracts, all of which contain abundant proteinase activity, in which no intraperitoneal sensitization phase is required.

8. With the realization that asthma and related disorders are related to airway fungal infections¹ and that allergic inflammation is an elaborate antifungal program^{2, 3}, there is concern that inhibition of caspase 4/11, as proposed to treat asthma, might compromise antifungal immunity and worsen disease in the long run. This concern is particularly apropos given that yeasts and molds can both activate caspase 11 and trigger pyroptosis^{4, 5}. Please temper discussion points accordingly.

Other:

1. Introduction first paragraph, last sentence: of caspase -11 (not od)
2. Figure 6E is cited in the text twice, but the second citation should be for Figure 6F.

References

1. Porter PC, et al. Airway surface mycosis in chronic TH2-associated airway disease. *J Allergy Clin Immunol* 134, 325-331 (2014).
2. Porter P, et al. Necessary and sufficient role for T helper cells to prevent fungal dissemination during mucosal airway infection. *Infect Immun* 79, 4459-4471 (2011).
3. Porter P, et al. Link between allergic asthma and airway mucosal infection suggested by proteinase-secreting household fungi. *Mucosal Immunol* 2, 504-517 (2009).
4. Sun Y, et al. Neutrophil Caspase-11 Is Required for Cleavage of Caspase-1 and Secretion of IL-1beta in *Aspergillus fumigatus* Infection. *Journal of Immunology* 201, 2767-2775 (2018).
5. Gabrielli E, et al. Induction of caspase-11 by aspartyl proteinases of *Candida albicans* and implication in promoting inflammatory response. *Infection & Immunity* 83, 1940-1948 (2015).

David B. Corry, M.D.

Reviewer #3 (Caspase, inflammasome)(Remarks to the Author):

Activation of murine caspase-11 or human caspase-4 by intracellular LPS induces an inflammatory cell death called pyroptosis, an important host defense response to fight against pathogen infection. Excessive activation of caspase-11 causes sepsis or other inflammatory disorders. In the submitted manuscript, Zaslona et al. report that Prostaglandin E2 (PGE2) treatment inhibits LPS-induced transcription of caspase-11 through downregulation of IFN β expression. In addition, as addition of PGE2 post-LPS priming can also inhibit LPS transfection-induced pyroptosis, they further propose that PGE2 can induce Ser-350 phosphorylation of caspase-11, which is inhibitory for the function of caspase-11 to trigger pyroptosis (presumably through cleavage and activation of Gasdermin-D). Lastly, the authors show that expression of caspase-11 increases in a mouse model of asthma, which is sensitive to inhibition by PGE2, and strikingly caspase-11 knockout mice are resistant to asthma development. Based on these results, they put forward an interesting hypothesis that PGE2 has an important function of protecting the mice or maybe also patients from asthma. There are quite a bit of interesting and important observations in this manuscript that need to be published soon. The weakest part of the study is PGE2-induced Ser-350 phosphorylation of caspase-11 that serves as an inhibitory mechanism, which is not convincing to this reviewer.

Major issues:

1. In Fig. 1D, EP2-deficient BMDMs show a higher expression of caspase-11. Are the cells more sensitive to LPS transfection-induced pyroptosis in the absence of priming, which then should be resistant to inhibition by PGE2?
2. Fig. 1F shows that although LPS priming increases expression of caspase-4, non-primed human PBMCs do express caspase-4. If that is the case, why LPS transfection did not stimulate pyroptosis of PBMCs in the absence of priming, which is what Fig. 1E shows?
3. Fig. 2D should include a control, i.e. the effect of PGE2 on IFN β -induced binding of STAT1 to the caspase-11 promoter, which is a perfect one to strengthen the connection there.
4. The biggest issue in this manuscript is Fig. 4 which fails to demonstrate that PGE2 indeed induces Ser-350 phosphorylation in caspase-11 to inhibit the caspase function. The mass spec data are not convincing as there are no tandem mass spec spectra and there are no quantifications. The authors should make a stable cell line expressing the catalytic cysteine mutant (to block autoprocessing) of a tagged version of caspase-11. With that, the authors can then immuno-purify caspase-11 (PGE2-treated or untreated) and perform mass spec analyses as well as phos-tag gel analyses. The S350A mutant can also be included to confirm the observation. Also, if these assay work well, the PKA agonist can be tested to confirm the PGE2 indeed acts through PKA.
5. How does Ser-350 phosphorylation inhibit caspase-11 function? Where is the residue located in the structure of caspase-11? Does the phosphorylation inhibit caspase-11 proteolytic activity? Is the phospho-mimic mutant resistant to PGE2 treatment? Also, the difference between caspase-11 WT and S350A appears to be very subtle, not in agreement with the cell death data in Fig. 3.
6. Fig. 6 data are nice and strong in demonstrating that the presence of caspase-11 drives mouse asthma pathology in the OVA model with a consistent immunological mechanism. However, the authors did not directly show the inhibitory effect of PGE2 but only referred previous publications. It would be nice to have a complete and cohesive story if the authors can show the effect of PGE2 in this model and also test whether the EP2 knockout mice are more sensitive in developing asthma, which resists the treatment of PGE2 and can be blocked in the EP2 and caspase-11 double KO mice.

Minor issues:

1. There is a typo in the last sentence of the first paragraph of the Introduction, od should be of.
2. In the last sentence of the first paragraph in Results session, it is better to use "downregulate" rather than "block" as PGE2 does not inhibit pyroptosis completely.
3. Real-time PCR data should be shown for Fig. 2B so that the data are cohesive and in the same format with those shown in Fig. 2A.
4. In Fig 3E, why does the input show no priming effect on caspase-11 expression, which is inconsistent with the data in Fig. 3D?
5. What are the two bands in the Fig. 4C blot? Caspase-11 or Gasdermin-D or both?
6. The legend in Fig. 5A indicates probing of Gasdermin-D expression, but the data did show that. What is the numbering in Fig. 5A and 5C, different mice? If so, N should be 4 rather than 6 in Fig. 5B. Also, why does the densitometry of caspase-11 expression (Y-axis) differs in Fig 5B and 5D?

Reviewer #1 (Inflammasome, cell death)(Remarks to the Author):

Zaslona et al. explore the role of the anti-inflammatory prostaglandin PGE₂ in regulating the inflammatory caspase-11 and the role of caspase-11 in asthma. Using in vitro assays, they demonstrate the PGE₂ decreases the ability of LPS priming to induce caspase-11 expression likely by downregulating TLR4 driven IFN γ production. Additionally, they argue that PGE₂ causes phosphorylation of caspase-11 at serine-350, shown by mass spectrometry. They argue that this is a second way that PGE₂ may inhibit caspase-11, explored using transfection of mutant caspase-11. Furthermore, they observed that their caspase-11^{-/-} mice had decreased levels of various immunologic parameters in the lungs of mice in an asthma model.

Ultimately, the largest weakness of this paper is a lack of significant impact/novelty, which may be partly due to the fact that this paper reads as two small papers strung together into a single paper whereby the first story (regulation of caspase-11 by PGE₂) seems only peripherally connected to the second story (caspase-11 in asthma) by the observation that PGE₂ helps in asthma. The conclusion that PGE₂ inhibits caspase-11 expression by inhibiting autocrine IFN γ is expected from published literature (PGE₂ is known to inhibit IFN γ expression and IFN γ expression is known to drive caspase-11 expression already). Conversely, the significance of phosphorylation of caspase-11 needs to be more firmly demonstrated. Finally, while clearly establishing a role for caspase-11 in asthma would be significant, more of this particular story would need to be established (eg how is caspase-11 activated) and issues of potential confounders (lung microbiota, potential LPS contamination, etc) would need to be more clearly addressed to make the asthma data publishable. In summary, this manuscript is not suited for publication at all, and an immense amount of work remains to be done.

Major points:

There seems to be no hypothesis that forms the premise of this paper. Why did the authors think that PGE₂ would inhibit caspase-11 in the first place?

As we stated in the introduction, PGE₂ is a well-known macrophage regulator, mostly inhibiting macrophage function and more globally facilitating tissue repair and a return to homeostasis. IL-1 β is a prominent proinflammatory cytokine, produced in response to activation of the canonical (caspase-1) or non-canonical (caspase-11) inflammasomes, both of which lead to pyroptosis. While previous studies have explored the effect of PGE₂ on canonical inflammasome activation (Sokolowska, M. et al, 2015. Prostaglandin E₂ Inhibits NLRP3 Inflammasome Activation through EP4 Receptor and Intracellular

Cyclic AMP in Human Macrophages. *J Immunol* 194: 5472-5487; Mortimer, L., et al 2016. NLRP3 inflammasome inhibition is disrupted in a group of auto-inflammatory disease CAPS mutations. *Nat Immunol* 17: 1176-1186.) the effect of PGE₂ on non-canonical inflammasome activation (which would promote tissue injury via cell death) is unexplored. We therefore thought it would be of substantial interest to examine whether PGE₂ might limit this process as part of its mechanism in the prevention of tissue injury. This was the rationale for our study, since if we found that PGE₂ might limit caspase-11, this would further support caspase-11 as an important mediator of tissue injury and provide new mechanistic insights into how PGE₂ is cytoprotective. In the revised version of the manuscript we have explained this more clearly (page 3).

We would also point out that we chose asthma/allergic airway inflammation as an *in vivo* model because it is a disease where PGE₂ has a protective effect and of direct relevance here, because pyroptosis has been implicated in asthma (Panganiban R.A. et al A functional splice variant associated with decreased asthma risk abolishes the ability of gasdermin B to induce epithelial cell pyroptosis. 2018 *J Allergy Clin Immunol* 142: 1469-1478 e1462). Furthermore, the gene encoding Gasdermin B has been shown to be strongly associated with asthma (Moffatt M.F. et al, A large-scale, consortium-based genomewide association study of asthma. 2010 *N Engl J Med* 363: 1211-1221) and a recent study has shown that Gasdermin B promotes Caspase-4 activity (Chen Q et al, GSDMB promotes non-canonical pyroptosis by enhancing caspase-4 activity. *J Mol Cell Biol.* 2018 Oct 15. doi: 10.1093/jmcb/mjy056.). Our experiments support a critical role for caspase-11 (caspase-4 in human) in asthma/allergic airway inflammation, not only from the caspase-11-deficient mice being protected in the model we use, but also because PGE₂ (which protects against asthma) inhibits caspase-11. The two parts of our study are therefore complimentary and support each other: PGE₂ inhibiting caspase-11 (via 2 separate processes), with this inhibitory effect being validated in an *in vivo* model of asthma/allergic inflammation. Our results with PGE₂ therefore not only point to a mechanism whereby PGE₂ prevents tissue injury but also supports the role of caspase-11/4 in asthma. We have modified the revised version of the manuscript in order to integrate both aspects and keep the story coherent (page 4).

We have known at least since 2008 that PGE2 inhibits TLR4 driven type I IFN production. Since caspase-11 is dependent upon this, it is not novel to show the link again. However, this part of the paper could be published as a PLOS One paper, verifying expected results.

Prostaglandin E2 Suppresses Lipopolysaccharide-Stimulated IFN- β Production

X. Julia Xu, Jonathan S. Reichner, Balduino Mastrofrancesco, William L. Henry, Jr. and Jorge E. Albina J Immunol 2008; 180:2125-2131doi: 10.4049/jimmunol.180.4.2125

We agree with the reviewer that the inhibition of type I interferon is not novel, and we cite the paper mentioned. However, we feel that Inhibition of caspase-11 expression by PGE₂ (although expected because of the effect of PGE₂ on type I interferon production) is a novel finding, especially in the context of the additional effect of PGE₂ on caspase-11 activity. Both findings emphasize the likely importance of the targeting of caspase-11 by PGE₂.

Fig. 3, the cells are still being exposed overnight to TLR4 stimulating LPS in the context of overnight PGE2, so this really cannot rule out that TLR4 signaling is being affected by the PGE2 continuously overnight.

In experiments in Figure 3 PGE₂ was added for 30 minutes only. The medium was removed and replaced with new medium containing LPS encapsulated in liposomes, which was incubated overnight. Therefore, there was no PGE₂ in the overnight culture. Moreover, we have shown Western Blot analysis of caspase-11 expression, which remained unchanged in these experimental conditions, indicating that ongoing inhibition of TLR4 overnight was unlikely to be occurring, since this would be expected to decrease caspase-11 expression. We have clarified the experimental setup in figure legend and emphasized this point (page 6 and 30) to avoid any misunderstanding.

Fig. 3 shows that PGE2 causes detectable caspase-11 phosphorylation, however this may not have any physiologic relevance. It may be only 1% of the available caspase-11 protein pool that is phosphorylated, which often happens with kinases, and this may no real importance.

In the revised version of the manuscript we have included new data (Figure 4A) from

primary bone marrow derived macrophages and report phosphorylation detected using a phos-tag mobility shift gel in cells treated with PGE₂. We also present new data for serine phosphorylation of overexpressed caspase-11 in response to PGE₂ in Fig. 4B. These data are described on page 7. Although non-quantitative, we feel that based on Western Blotting these changes are substantial. We provide evidence that Serine 350 phosphorylation is relevant in caspase-11 activity as shown in Figure 4F, where an S350A mutant (but importantly not the other serine mutants) is a more potent inducer of cell death than wild type caspase-11. These data are described on page 7. Finally, as shown in Figure 4G and described on page 8, using structural modeling we have determined that Serine 350 is located on the cell surface in the intrinsically disordered region (IDR), which for serines in other proteins has been shown to be phosphorylated resulting in the inhibition of protein activity (Kasahara K et al Phosphorylation of an intrinsically disordered region of Ets1 shifts a multi-modal interaction ensemble to an auto-inhibitory state. *Nucleic Acids Res.* 2018 Mar 16; 46(5): 2243–2251). These data are described on page 8. Overall, we believe our data is valuable because no posttranslational modification of caspase-11 has been identified so far and our data suggest functional relevance for S350phosphorylation.

Regarding the 293 overexpressed point mutants in Fig. 3, while their data suggest a potential role of phosphorylation, they do not clearly demonstrate that phosphorylation as opposed to some other issue (eg difference in folding affecting the auto-activation properties of the protein when the protein is overexpressed) as the cause of the observed mild difference in LDH release. There is no treatment with PGE₂ here, and there is no activation of caspase-11 by LPS. Thus these experiments are quite non-physiologic.

We struggled to obtain enough material for mass spectrometry for phosphorylation analysis using primary macrophages or the RAW cell line, or to transfect them sufficiently with the point mutants to examine for cell death, so we resorted to 293T cells. We obtained enough material for mass spectrometry analysis and identified the phosphorylation sites in PGE₂ – treated 293T cells. For the cell death assay, although we have not used LPS to stimulate 293T cells, as the reviewer mentioned

overexpressed caspase-11 will undergo autoactivation and subsequent pyroptosis similar to LPS transfection. As suggested by the reviewer, we attempted to inhibit pyroptosis in the 293T cell over-expressing caspase-11 but we were unable to demonstrate inhibition. A possible explanation is the inability of PGE₂ to overcome the strength of the pyroptotic signal elicited by the over-expressed caspase-11.

In figure 3A, why is % cell death of cells so much less than in figure 1A LPS-transfected, PGE₂ untreated controls (~20% vs ~80%) despite what appears to be only minor variation in protocol?

The difference in cell death is likely to have been due to the protocol where media was not changed in Fig. 1A. PGE₂ and control were added followed by LPS priming for 4h followed by LPS transfection with cell death measured after overnight incubation. In Fig. 3A cells were primed with LPS for 4h, media was removed (as stated in the revised version of the figure description on page 30) followed by LPS transfection with cell death measured after overnight incubation. The first protocol results in a higher level of basal cell death. However, the fold increase in cell death with transfected LPS (and fold inhibition by PGE₂) is very similar between protocols. LPS priming may be cytoprotective or the media change is likely to remove some of the LDH being produced basally.

While the authors demonstrate a difference in immunologic parameters within the lung in Fig. 4 and one would expect that these parameters correlate with asthma phenotype, the authors should more directly demonstrate an effect on airway hypersensitivity (eg methacholine challenge). One wants to actually show that the gene affects the clinical phenotype.

Although our OVA model is commonly known as a murine asthma model, as reviewer 2 pointed out it is more a model of allergic lung inflammation. Nonetheless it is an acute model with immune cell infiltration, but very limited airway remodeling which is more pronounced in more chronic models, where methacholine challenge will provoke bronchoconstriction. In our manuscript we rely on eosinophil counts as an asthma marker, since there is a strong correlation between eosinophil count and methacholine challenge (Schwartz et al Correlation between eosinophil count and methacholine

challenge test in asymptomatic subjects. *J Asthma*. 2012 May;49(4):336-41). Our results describe lung inflammation in detail, and specifically eosinophil count and IL-5 levels, which have very strong clinical associations with asthma and are targeted therapeutically (Nair P et al Oral Glucocorticoid-Sparing Effect of Benralizumab in Severe Asthma. *N Engl J Med*. 2017 Jun 22;376(25):2448-2458) and (Ortega H.G. Mepolizumab treatment in patients with severe eosinophilic asthma. *N Engl J Med*. 2014 Sep 25;371(13):1198-207). Finally, we confirm the importance of caspase-4 in asthma by showing increased expression in lung samples from patients (Fig. 6H). We hope that this will suffice for the reviewer.

The idea that endogenous lipids activates caspase-11 (reference 5) is extremely controversial, and cannot be taken as a given for this study. Thus, it remains unclear how caspase-11 would be activated in the asthma model, where is the LPS?

The lung is not as sterile an environment as we once thought. It is very likely that once damage occurs to lung epithelial cells Gram negative bacteria might activate Caspase-11 and cause inflammation contributing to the asthma phenotype. In the revised version of the manuscript on page 12 we have discussed that LPS has been demonstrated to worsen asthma symptoms, which could in part be due to caspase-11 activation.

There have been conflicting evidence regarding the roles of inflammasomes and downstream IL-1b in mouse models of asthma. In light of this fact and the fact that the mechanism by which caspase-11 is activated in this model is not addressed, the authors need to address potential confounders such as contributory effects of differing lung microbiota. They need to use littermate controlled mice – this is essential, very important.

We have used littermate-controlled mice in all of our studies and have now stated this clearly in the materials and methods section.

They also need to consider the possibility of LPS contamination of reagents.

We agree that we cannot state that our model is 100% LPS free. We have discussed the possibility that LPS in Ovalbumin might contribute to caspase-11 expression in our study on page 12.

Figure 5 is not definitive that the PGE2 effect on caspase-11 is occurring in the asthma model. It is only correlative. This drives home that there is really a disconnect between the two stories: the one that PGE2 affects caspase-11, and the other that caspase-11 is involved in asthma.

We feel that it is appropriate to present both parts of our study together. We have presented an *in vivo* model of asthma where Caspase-11 is induced. Upon exogenous administration of PGE₂ in the model caspase-11 expression is inhibited and when cyclooxygenase is inhibited *in vivo* with indomethacin, caspase-11 expression is further induced. These data support *in vivo* regulation of Caspase-11 by PGE₂. Since caspase-11 - deficient mice are protected in this model and given the *in vivo* inhibition of its induction by PGE₂ (and its enhanced expression by inhibition of cyclooxygenase) we feel it is reasonable to conclude that inhibition of caspase-11 by PGE₂ is part of its protective effect here, and that this aspect further validates the role of caspase-11 in asthma.

Minor points.

Introduction stating that PGE2 is the primary target of NSAIDs. I don't think that is correct, since NSAIDs inhibit COX.

We have modified the manuscript describing NSAIDs as COX1 and COX2 inhibitors, which results in inhibition of PGE₂ as well as other prostanoids.

Fig. 2 is explained incorrectly. the STAT1 phosphorylation is downstream of IFN- β . Thus, the inhibition of IFN- β by PGE2 explains the lower phospho STAT1 in Fig. 2C-D, which is not the impression given by the text.

We have modified text in the revised version of the manuscript, so it is evident that the

observed decrease in STAT-1 phosphorylation is caused by inhibition of IFN- β production by PGE₂.

We thank the reviewer for raising these issues and hope that our response will be satisfactory. We feel we have a compelling finding here describing how PGE₂ as an endogenous regulator of tissue injury and inflammation is inhibiting caspase-11, a key player in cell death and inflammation. Our work has a particular relevance to asthma pathogenesis, where both PGE₂ (as a protector) and caspase-11 (as a driver of disease - from our own study here and supported by others more indirectly) have been analyzed. Our work helps explain why PGE₂ is protective in asthma via an inhibitory effect on caspase-11, further implicating the targeting of caspase-11 as a cytoprotective mechanism for PGE₂.

Reviewer #2 (Airway inflammation, Th function)(Remarks to the Author):

The investigators have evaluated the role of caspase -11 in the expression of allergic airway disease and the inhibitory effect of PGE2. The data are excellent, as far as they go, and the message is very interesting and important, but there are significant issues to overcome at the same time.

We thank the reviewer for these positive comments.

1. Western blots: densitometric quantification is helpfully employed for some westerns. Please expand use to all such blots to enhance understanding of the semiquantitative changes.

We have included quantification values in the graphs.

2. Figure 5. It is confusing to label conditions as “asthma” when this term is only properly used in the context of human disease. Suggest referring to mouse disease as “allergic airway disease” (AAD) or something similar to avoid confusion.

We agree with the reviewer that asthma is not an accurate description. We have re-labeled mouse experiments to AAD and explained in the text that this model aims to mimic human asthma.

3. Figure 5A: there are doublets for IL-1b and cas11-please add arrows to indicate what are the true bands of interest.

The Caspase-11 bands isoforms and were taken into consideration for densitometric analysis. The caspase-11 locus encodes two proteins of 38 and 43 kD (Wang S et al Identification and characterization of Ich-3, a member of the interleukin-1beta converting enzyme (ICE)/Ced-3 family and an upstream regulator of ICE. J Biol Chem. 1996 Aug 23;271(34):20580-7).

4. Please describe what a “lung lysate” is and how they are prepared.

After mice were sacrificed whole lung lobes were snap frozen in liquid nitrogen and stored in -80 for further analysis. On the day of isolation lungs were homogenized in RIPA buffer in the presence of a cocktail of protease inhibitors and subsequently spun down. Supernatant was mixed with sample buffer and subjected to WB analysis. We have included this information in the materials and methods section.

5. The authors exaggerate by saying that in the absence of caspase 11 an adaptive immune response is not mounted-it is reduced, but not abrogated.

We have changed the wording. We have clarified that although systemically the immune responses in caspase-11 deficient mice were reduced, the infiltration of immune effector cells to the lung was indeed abrogated. CD45 staining, which we used to discriminate between tissue resident and recruited inflammatory cells demonstrated that locally in

the lung there was no infiltration. These data together with histological evidence show full lung protection from early allergic inflammation in lungs of caspase-11 deficient mice.

6. Allergic airway disease model: this is not an asthma model as there are no measurements of airway mechanics, e.g., airway responsiveness to methacholine or acetylcholine challenge. Please add these data if these measurements are possible. If the data cannot be obtained, the model should be referred to as something else, such as "model of eosinophilic lung inflammation, etc.

We have changed the name of the model to allergic airway disease (AAD) and cite work which shows a strong correlation between lung inflammation and lung function in our model. Strong correlation between eosinophil count and methacholine challenge has been demonstrated previously (Schwartz N et al Correlation between eosinophil count and methacholine challenge test in asymptomatic subjects. *J Asthma*. 2012 May;49(4):336-41; Lee Y.J. et al Association between eosinophilic airway inflammation and persistent airflow limitation. *J Asthma*. 2013 May;50(4):342-6.). Since we do have asthmatic human data showing increase in caspase-4 we believe discussing our results in the context of asthma is reasonable.

*7. The ovalbumin model has faded in relevance in recent years with the gradual understanding that asthma and related diseases are proteinase-activated diseases. More confidence in the in vivo data would be possible by repeating experiments in which the allergic airway disease was induced by a proteinase allergen (e.g., proteinase of *Aspergillus oryzae* or *A. melleus*), or a whole fungal, whole dust mite, or whole pollen extracts, all of which contain abundant proteinase activity, in which no intraperitoneal sensitization phase is required.*

We agree with the reviewer that these models are more relevant as asthma model specifically with the aspect of lung sensitization. In the revised version of the manuscript we cite work demonstrating the importance of pyroptosis as a process in asthma (Panganiban R.A. et al A functional splice variant associated with decreased asthma risk abolishes the ability of gasdermin B to induce epithelial cell pyroptosis. 2018 *J Allergy Clin Immunol* 142: 1469-1478 e1462.). We also cite work presenting strong association of the gene encoding Gasdermin B (which is involved in pyroptosis) in

asthma pathogenesis (Moffatt M.F. et al A large-scale, consortium-based genomewide association study of asthma. 2010 N Engl J Med 363: 1211-1221.) and a very recent study showing that Gasdermin B promotes caspase-4 activity (Chen Q. et al GSDMB promotes non-canonical pyroptosis by enhancing caspase-4 activity. J Mol Cell Biol. 2018 Oct 15. doi: 10.1093/jmcb/mjy056.). Moreover, we present data in Fig 6H demonstrating increased expression of caspase-4 protein in actual human asthmatics. So, although we agree our in vivo model is a model of allergic airway disease, we believe we provide a strong case for the importance of caspase-4 in asthma.

8. With the realization that asthma and related disorders are related to airway fungal infections¹ and that allergic inflammation is an elaborate antifungal program^{2, 3}, there is concern that inhibition of caspase 4/11, as proposed to treat asthma, might compromise antifungal immunity and worsen disease in the long run. This concern is particularly apropos given that yeasts and molds can both activate caspase 11 and trigger pyroptosis^{4, 5}. Please temper discussion points accordingly.

We agree with the reviewer and as is often the case with inhibition of targets which dampen inflammation these can worsen host defense. We have discussed this issue on page 13 of the discussion.

Other:

- 1. Introduction first paragraph, last sentence: of caspase -11 (not od)*
- 2. Figure 6E is cited in the text twice, but the second citation should be for Figure 6F.*

These have been corrected.

References

- 1. Porter PC, et al. Airway surface mycosis in chronic TH2-associated airway disease. J Allergy Clin Immunol 134, 325-331 (2014).*
- 2. Porter P, et al. Necessary and sufficient role for T helper cells to prevent fungal dissemination during mucosal airway infection. Infect Immun 79, 4459-4471 (2011).*

3. Porter P, et al. Link between allergic asthma and airway mucosal infection suggested by proteinase-secreting household fungi. *Mucosal Immunol* 2, 504-517 (2009).

4. Sun Y, et al. Neutrophil Caspase-11 Is Required for Cleavage of Caspase-1 and Secretion of IL-1beta in *Aspergillus fumigatus* Infection. *Journal of Immunology* 201, 2767-2775 (2018).

5. Gabrielli E, et al. Induction of caspase-11 by aspartyl proteinases of *Candida albicans* and implication in promoting inflammatory response. *Infection & Immunity* 83, 1940-1948 (2015).

David B. Corry, M.D.

We thank the reviewer for the comments made, and we hope our responses will be deemed satisfactory.

Reviewer #3 (Caspase, inflammasome)(Remarks to the Author):

Activation of murine caspase-11 or human caspase-4 by intracellular LPS induces an inflammatory cell death called pyroptosis, an important host defense response to fight against pathogen infection. Excessive activation of caspase-11 causes sepsis or other inflammatory disorders. In the submitted manuscript, Zaslona et al. report that Prostaglandin E2 (PGE2) treatment inhibits LPS-induced transcription of caspase-11 through downregulation of IFN β expression. In addition, as addition of PGE2 post-LPS priming can also inhibit LPS transfection-induced pyroptosis, they further propose that PGE can induce Ser-350 phosphorylation of caspase-11, which is inhibitory for the function of caspase-11 to trigger pyroptosis (presumably through cleavage and activation of Gasdermin-D). Lastly, the authors show that expression of caspase-11 increases in a mouse model of asthma, which is sensitive to inhibition by PGE2, and strikingly caspase-11 knockout mice are resistant to asthma development.

Based on these results, they put forward an interesting hypothesis that PGE2 has an important function of protecting the mice or maybe also patients from asthma. There are quite a bit of interesting and important observations in this manuscript that need to be published soon. The weakest part of the study is PGE2-induced Ser-350 phosphorylation of caspase-11 that serves as an inhibitory mechanism, which is not convincing to this reviewer.

We thank the reviewer for stating that our hypothesis is interesting and important, and that our work should be published soon. As we describe below, we provide extra evidence for caspase-11 phosphorylation, which we hope will help address the concerns of the reviewer on this issue.

Major issues:

1. In Fig. 1D, EP2-deficient BMDMs show a higher expression of caspase-11. Are the cells more sensitive to LPS transfection-induced pyroptosis in the absence of priming, which then should be resistant to inhibition by PGE2?

This would be a very interesting experiment. However, sadly we no longer have access to these mice and organizing this work would not allow us to provide our revision in the suggested time frame. We therefore discuss this potential experiment as future work to be done on page 13.

2. Fig. 1F shows that although LPS priming increases expression of caspase-4, non-primed human PBMCs do express caspase-4. If that is the case, why LPS transfection did not stimulate pyroptosis of PBMCs in the absence of priming, which is what Fig. 1E shows?

Even though there is indeed expression of caspase-4 basally, in our hands we still have to prime with LPS. This could be due to there being insufficient caspase-4 in non-primed cells or perhaps other components being needed which LPS induces. We mention this on page 5.

3. Fig. 2D should include a control, i.e. the effect of PGE2 on IFN β -induced binding of STAT1 to the caspase-11 promoter, which is a perfect one to strengthen the connection there.

PGE₂ did not block the induction of caspase-11 by IFN-β as depicted in Fig. 2B, therefore we didn't include IFN-β in the STAT1 pulldown experiment. We have highlighted this finding in the revised version of the manuscript on page 6 and hope that the lack of effect of PGE₂ on caspase-11 induction by IFN-β will suffice as a control here.

4. The biggest issue in this manuscript is Fig. 4 which fails to demonstrate that PGE₂ indeed induces Ser-350 phosphorylation in caspase-11 to inhibit the caspase function. The mass spec data are not convincing as there are no tandem mass spec spectra and there are no quantifications. The authors should make a stable cell line expressing the catalytic cysteine mutant (to block autoprocessing) of a tagged version of caspase-11. With that, the authors can then immuno-purify caspase-11 (PGE₂-treated or untreated) and perform mass spec analyses as well as phos-tag gel analyses. The S350A mutant can also be included to confirm the observation. Also, if these assay work well, the PKA agonist can be tested to confirm the PGE₂ indeed acts through PKA.

We agree with the reviewer that this is a crucial point of the manuscript and we performed new experiments to strengthen our findings. Firstly, we have performed phos-tag gel analysis from primary BMDMs as suggested, and importantly we have observed a significant shift validating phosphorylation suggested by our mass spec data (Fig. 4A). Moreover, we used a phosphoserine-specific antibody on caspase-11 immunoprecipitates from 293T cells. We observed enhanced serine phosphorylation upon PGE₂ treatment (Fig. 4B). These data are described on page 7.

We made several attempts to analyze phosphorylation of catalytic cysteine mutant caspase-11, as well as the S350A mutant as suggested by the reviewer without success. Phosphorylation of the S350A mutant was still evident possibly due to the other serines being phosphorylated, as we had shown in the mass spec analysis that other serines could act as phosphoacceptors. Sadly, therefore this assay could not be deployed. However, given our data on caspase-11 phosphorylation as assessed by the phos-tag approach, phosphoserine immunoblotting, the mass spec identifying S350 as a phosphoacceptor from PGE₂ treated cells, the observation that only the S350A mutant caspase-11 (and not the other serine mutants) demonstrated enhanced cell death, and the modelling of caspase-11 indicating S350 as being on the surface in an intrinsically

disordered region, we hope that the reviewer will allow us to publish our findings indicating caspase-11 phosphorylation in response to PGE₂ as a negative signal.

5. *How does Ser-350 phosphorylation inhibit caspase-11 function? Where is the residue located in the structure of caspase-11? Does the phosphorylation inhibit caspase-11 proteolytic activity? Is the phospho-mimic mutant resistant to PGE₂ treatment? Also, the difference between caspase-11 WT and S350A appears to be very subtle, not in agreement with the cell death data in Fig. 3.*

We have used PHYRE2 - a protein modeling computer program based on publication by Kelley et al "The Phyre2 web portal for protein modeling, prediction and analysis." *Nature Protocols volume 10, pages 845–858 (2015)* and visualized using EzMol software. PHYRE2 predicted Ser 350 to be located in the intrinsically disordered region (IDR) of caspase-11. This region is often very important for interactions with other molecules and/or targets of PTMs for signal transduction. Sites of kinases are enriched in IDR regions and can result in inhibition of protein activity as shown by Kasahara K et al Phosphorylation of an intrinsically disordered region of Ets1 shifts a multi-modal interaction ensemble to an auto-inhibitory state. *Nucleic Acids Res. 2018 Mar 16; 46(5): 2243–2251.* We have included the modelling data in the new Fig. 4G and discussion of the possible significance of S350 in the biology of caspase-11 in the revised version of the manuscript on page 8. We are aware that phosphorylated S350 might not fully explain the inhibitory effect of PGE₂ on caspase-11 activity, therefore in the revised version of the manuscript we refer to S350 phosphorylation as a mechanism, which at least partially explains the effect of PGE₂.

6. *Fig. 6 data are nice and strong in demonstrating that the presence of caspase-11 drives mouse asthma pathology in the OVA model with a consistent immunological mechanism. However, the authors did not directly show the inhibitory effect of PGE₂ but only referred previous publications. It would be nice to have a complete and cohesive story if the authors can show the effect of PGE₂ in this model and also test whether the EP2 knockout mice are more sensitive in developing asthma, which resists the treatment of PGE₂ and can be blocked in the EP2 and caspase-11 double KO mice.*

We have done new experiments (new Fig. 5A-E) with misoprostol (a pharmacological mimic of PGE₂) and added evidence of inhibition in our asthma model. Just as for caspase-11 deficient mice the local lung inflammation was fully inhibited, while the systemic response was diminished by misoprostol treatment. We have discussed the new results in the revised version on page 8, of the manuscript. Unfortunately, we no longer have access to EP2 deficient mice.

Minor issues:

1. There is a typo in the last sentence of the first paragraph of the Introduction, od should be of.

We have corrected this typo

2. In the last sentence of the first paragraph in Results session, it is better to use “downregulate” rather than “block” as PGE2 does not inhibit pyroptosis completely.

We have corrected this overstatement

3. Real-time PCR data should be shown for Fig. 2B so that the data are cohesive and in the same format with those shown in Fig. 2A.

We prefer protein levels and have included densitometric analysis to make it more cohesive. We believe that data in Fig. 2A leads to 2B.

4. In Fig 3E, why does the input show no priming effect on caspase-11 expression, which is inconsistent with the data in Fig. 3D?

We agree that control cells in Fig. 3E should not have any expression of caspase-11 at baseline and that LPS should have boosted caspase-11 expression. However, we

believe that this does not change our conclusions, as there was a sufficient amount of caspase-11 after priming to address whether PGE₂ could inhibit LPS binding.

5. What are the two bands in the Fig. 4C blot? Caspase-11 or Gasdermin-D or both?

Lane 2 has caspase-11 alone, and the upper band is caspase-11 (the faster migrating band is a non-specific band). Equal expression of the caspase-11 mutants can be seen in lanes 3 – 7. The uppermost band in these lanes is Gasdermin D (which like caspase-11 is FLAG tagged), and it is likely that the fastest migrating bands are cleavage products of Gasdermin D.

6. The legend in Fig. 5A indicates probing of Gasdermin-D expression, but the data did show that. What is the numbering in Fig. 5A and 5C, different mice? If so, N should be 4 rather than 6 in Fig. 5B. Also, why does the densitometry of caspase-11 expression (Y-axis) differs in Fig 5B and 5D?

We are sorry for this mistake; indeed, we have only assessed Caspase-11 and IL1- β . Numbers indicate different animals. Densitometry in 5B was calculated as fold change where control was set as 1 and in 5D was relative to caspase-11. We apologize for not making it consistent. In the revised version we have shown both as fold change.

We thank the reviewer for the comments and hope that we have addressed them satisfactorily. We have tried hard to provide additional evidence for caspase-11 phosphorylation and feel that the extra data strengthens our conclusions.

Reviewers' comments:

Reviewer #1 (Remarks to the Author):

Overall, the authors did not sufficiently address my comments. Where they did address them, the new data actually makes me more skeptical of the physiologic relevance of phosphorylation (new Fig. 4a). This manuscript remains two separate stories, one on PGE2 and the other on caspase-11 in asthma. Both stories require much more study prior to publication before they would be considered physiologically relevant. There remains a possibility that the two stories are connected, but the data as presented is not sufficient to draw this conclusion.

Major comments

The authors now state that the *in vivo* experiments used littermate controls, but that they forgot to include this in their original manuscript. I was surprised by this omission, since doing littermate controlled experiments requires a lot of planning, which tends to make one not forget to mention this important control. Typically, littermate controlled experiments are done by breeding heterozygous x homozygous knockout mice, yielding 50% heterozygous Casp11^{+/-} and 50% knockout Casp11^{-/-} mice. However, the authors have labelled their mice as "wild type", which is the label that would be used for Casp11^{+/+} mice. Did the authors forget to label their mice as heterozygous? If not this raises another concern, because the only way to obtain Casp11^{+/+} and Casp11^{-/-} littermate controls is by heterozygous x heterozygous breeding, yielding 25% WT and 25% KO mice, with 50% het mice that are discarded. In this case, the authors are getting very few mice from each breeder cage. With numbers of mice as low as 4-5 mice per group throughout Figure 4, this raises the risk that there are cage effects. By random chance one could obtain WT mice from one breeder and KO mice from another. Thus, the authors should include a supplemental table that shows which breeder cages the WT and KO mice came from.

Authors cite reference 16 as part of a rationale to study caspase-11 in asthma. This reference is not a widely accepted paper in the field. Further, the link to gasdermin B to caspase-11 is even more tenuous as mice do not encode gasdermin B. Thus, the overall rationale for the study remains quite weak. This decreases the impact of the paper, because in the end it does have a strong foundation.

Figure 3 I previously raised the caveat that the PGE2 could be acting on the TLR4 response. The authors claim that PGE2 was only added for 30 minutes at the start of the experiment, then removed. However, it is unclear how long the PGE2 effect could last, so if it were added first, then LPS was added, PGE2 could affect TLR4 signaling still. Caspase-11 transcription is only one part of the priming event, the field also believes there is post translational priming events whose mechanisms remain unclear. Both can be mediated by TLR4. To rule out that PGE2 was acting on caspase-11 and not TLR4 in figure 3 would require TLR4 knockouts. Liposomal LPS will never be fully contained in liposomes and sequestered from TLR4.

I previously suggested that only a small fraction of caspase-11 in the cell might be phosphorylated, and that this could happen by off target phosphorylation by a kinase. If only 1% of caspase-11 were phosphorylated to inhibit it, then this would have no physiologic impact on caspase-11 overall. In new data in Fig. 4a the authors show that this is likely true. Although the authors would like to focus on the phosphorylated band, they need to also consider the non-phosphorylated band. Non-phosphorylated caspase-11 band does not change in its intensity. Therefore, probably 99% of the caspase-11 in the cell remains not phosphorylated. This argues directly against the authors model.

Also, apparently all of the caspase-11 is phosphorylated in the untreated state without PGE2 and/or LPS, and there is almost no expression of caspase-11 in the untreated state. This seems inconsistent with the published literature, where there is usually a decent resting caspase-11 expression.

Concern about the 293 overexpressed point mutants in Figure 4 was not addressed. The authors state that the experiment did not work, which is consistent with the concern that phosphorylation is not physiologically relevant. This is particularly important because it is the only experiment that is in the paper that tries to suggest that the phosphorylation has a functional effect.

New Figure 4b is not publication quality data.

I previously questioned the lack of experimental reproducibility with 20% vs 80% cell death in different experiments. The authors provide a guess as to why this might be, given a subtle change in protocol. However, they do not do any experiments to show that they can get a strong phenotype that is consistent within their manuscript from figure to figure to address the critique.

The authors' explanation as to how caspase-11 could be activated during asthma models is not sufficient and there is no data in the paper to support it. While the lung is not sterile, it is quite clean. There is not a copious amount of LPS in the lung as there is in the gut. This raises the risk of some artifact causing the phenotypes seen.

The in vivo model still remains correlative for caspase-11 and PGE2. This again raises the risk that there is not a direct link between caspase-11 and PGE2 in this model. This is the missing link that I previously criticized. These are two stories that do not have a causal link between them.

Reviewer #2 (Remarks to the Author):

The authors have responded to many previously raised issues, significantly improving the story. Without airway mechanics testing, the data remain less than fully compelling, but nonetheless very interesting.

Reviewer #3

(No specific comments for the AU, but only private remarks to the editor)

Our specific responses to the referee comments are as follows:

Reviewers' comments:

Reviewer #1 (Remarks to the Author):

'Overall, the authors did not sufficiently address my comments. Where they did address them, the new data actually makes me more skeptical of the physiologic relevance of phosphorylation (new Fig. 4a). This manuscript remains two separate stories, one on PGE₂ and the other on caspase-11 in asthma. Both stories require much more study prior to publication before they would be considered physiologically relevant. There remains a possibility that the two stories are connected, but the data as presented is not sufficient to draw this conclusion.'

Response: we understand the reviewers concerns but would ask that the reviewer show some leniency on this point. We re-emphasize that we feel that our work makes a significant contribution to the field of pyroptosis since ours is the first description of a role for caspase-11 in allergic asthma. The additional insights being provided by PGE₂ inhibiting caspase-11 (both via inhibition of Type I interferon production (which we still feel is an important finding here) and via phosphorylation of caspase-11 (again a wholly novel finding which we feel compliments the other extensive aspects in our manuscript) are important validators of caspase-11 in allergic asthma. This finding will also have broader interest, since PGE₂ is cytoprotective in multiple contexts. We therefore feel that the 2 aspects here, namely caspase-11 in allergic asthma and it's targeting by PGE₂, are linked, since not only do we find a role for caspase-11 in allergic asthma but also that PGE₂, a known protector in asthma, can target Caspase-11. We feel that the 2 aspects strengthen the case for Caspase-11/4 as a key driver of allergic asthma, the overall conclusion from our work.

'The authors now state that the in vivo experiments used littermate controls, but that they forgot to include this in their original manuscript.'

Response: This was indeed an error. We omitted the information stating that we used littermate controls in the original manuscript, however we can assure the reviewer that Caspase-11 mice in our animal facility are littermate controls. After genotyping, homozygous offspring (+/+ and -/-) were used in the experiments while heterozygous offspring (+/-) were sacrificed. In our manuscript +/+ offspring are labelled WT and -/- offspring are labelled as KO. This has been clarified in the methods section on page 15.

'Authors cite reference 16 as part of a rationale to study caspase-11 in asthma. This reference is not a widely accepted paper in the field. Further, the link to gasdermin B to caspase-11 is even more tenuous as mice do not encode gasdermin B. Thus, the overall rationale for the study remains quite weak. This decreases the impact of the paper, because in the end it does have a strong foundation.'

Response: The rationale for our study was to test whether pyroptosis would be a key process in allergic asthma. This reference was used to strengthen the overall rationale for the role of pyroptosis in asthma. Our data indeed indicate a role for pyroptosis in allergic asthma, including expression of Caspase-4 in samples from asthmatic patients.

'Figure 3 I previously raised the caveat that the PGE₂ could be acting on the TLR4 response. The authors claim that PGE₂ was only added for 30 minutes at the start of the experiment, then removed. However, it is unclear how long the PGE₂ effect could last, so if it were added first, then LPS was added, PGE₂ could affect TLR4 signaling still. Caspase-11 transcription is only one part of the priming event, the field also believes there is post translational priming events whose mechanisms remain unclear. Both can

be mediated by TLR4. To rule out that PGE₂ was acting on caspase-11 and not TLR4 in figure 3 would require TLR4 knockouts. Liposomal LPS will never be fully contained in liposomes and sequestered from TLR4.'

Response: In these experiments caspase-11 was already transcribed and translated when the PGE₂ was added so any new transcription which might have occurred after transfection indeed might have been affected by PGE₂. However, we re-emphasize that as shown in Fig 3A, when PGE₂ is added after LPS priming there is no effect on Caspase-11 protein levels, even though pyroptosis is clearly inhibited. We cannot exclude any other posttranslational modifications apart from phosphorylation which PGE₂ might be having on caspase-11 to affect its activity. We discuss this further in the revised version of the manuscript on page 6 and page 8.

'I previously suggested that only a small fraction of caspase-11 in the cell might be phosphorylated, and that this could happen by off target phosphorylation by a kinase. If only 1% of caspase-11 were phosphorylated to inhibit it, then this would have no physiologic impact on caspase-11 overall. In new data in Fig. 4a the authors show that this is likely true. Although the authors would like to focus on the phosphorylated band, they need to also consider the non-phosphorylated band. Non-phosphorylated caspase-11 band does not change in its intensity. Therefore, probably 99% of the caspase-11 in the cell remains not phosphorylated. This argues directly against the authors model.'

Response: Our experiments indicate that a substantial level of phosphorylation is occurring. In Figure 4A the phospho form is almost as intense as the non-phospho form. In Figure 4B we have added a new blot demonstrating a shift in Caspase-11 expression. Again, a significant proportion of Caspase-11 is shifted and the shifted form is shown to be phosphorylated on Serine. This is discussed on page 7.

'Also, apparently all of the caspase-11 is phosphorylated in the untreated state without PGE₂ and/or LPS, and there is almost no expression of caspase-11 in the untreated state. This seems inconsistent with the published literature, where there is usually a decent resting caspase-11 expression.'

Response: in our hands caspase-11 expression is negligible in the resting state (eg- Fig 1C and in subsequent figures). No phosphorylation is evident without PGE₂ treatment (as shown in Fig 4A-D, including by mass spectrometry).

'Concern about the 293 overexpressed point mutants in Figure 4 was not addressed. The authors state that the experiment did not work, which is consistent with the concern that phosphorylation is not physiologically relevant. This is particularly important because it is the only experiment that is in the paper that tries to suggest that the phosphorylation has a functional effect.'

Response: We have again tried to repeat the experiment in Figure 4E/F incorporating PGE₂. Titrating Caspase-11 proved difficult as low levels of plasmid did not cause any cell death. As we stated previously, this leads us to believe that although our data clearly show that PGE₂ inhibits Caspase-11-mediated pyroptosis in primary human and murine cells, the HEK-293T overexpression system is insensitive to PGE₂ inhibition. There could be multiple reasons for this including heightened PGE₂ breakdown. We have discussed this in the revised version of the manuscript on page 8. Although this experiment is therefore not providing additional evidence for Caspase-11 phosphorylation in response to PGE₂, we ask the referee to allow us to draw this overall conclusion, given the evidence we present on phospho-Caspase-11 in Figure 4A, on serine phosphorylation in Fig 4C/D (detection of serine-phosphorylated Caspase-11 by Western blotting) and Fig 4 C/D (demonstration of Caspase-11 phosphorylation on Serine 350 by mass spectrometry), and that this is likely to be inhibitory given the

increased activity of the Caspase-11 S350A mutant in Fig 4 E/F and the position of Serine 350 in the Caspase-11 structure shown in Fig 4G.

We ask the referee not to take this particular experiment in isolation from the rest of our study and again emphasize the other novel aspects of our work on the role of Caspase-11 in allergic asthma and its in vivo targeting by PGE₂, which also involves inhibition of interferon-beta production which blocks Caspase-11 induction. We feel these aspects of our study are equally important and novel.

'New Figure 4b is not publication quality data.'

Response: We have repeated these experiments. We include the new Fig.4 B where we have observed a shift of the Caspase-11 band upon PGE₂ treatment, the shifted band being identified as caspase-11 phosphorylated on serine. These data are described on page 7.

'I previously questioned the lack of experimental reproducibility with 20% vs 80% cell death in different experiments. The authors provide a guess as to why this might be, given a subtle change in protocol. However, they do not do any experiments to show that they can get a strong phenotype that is consistent within their manuscript from figure to figure to address the critique.'

Response: The percentage of cell death does vary between experiments but is always significant under the conditions shown, and importantly is significant enough to determine inhibitory effects by PGE₂ and other treatments here. We therefore feel that our data are robust enough for the conclusions being drawn. The range shown reflects differences in protocols and we discuss this on page 17.

'The authors' explanation as to how caspase-11 could be activated during asthma models is not sufficient and there is no data in the paper to support it. While the lung is not sterile, it is quite clean. There is not a copious amount of LPS in the lung as there is in the gut. This raises the risk of some artifact causing the phenotypes seen.'

Response: We do not know how Caspase-11 is being induced in the asthma model. LPS might be present in the airways or might even be coming from the gut microbiome. Endogenous factors from dying cells might also activate Caspase-11.

'The in vivo model still remains correlative for caspase-11 and PGE₂. This again raises the risk that there is not a direct link between caspase-11 and PGE₂ in this model. This is the missing link that I previously criticized. These are two stories that do not have a causal link between them.'

Response: We demonstrated that treatment of mice with the PGE₁ analogue misoprostol being protective in the in vivo model, whilst inhibiting cyclooxygenase increases expression of Caspase-11. We feel these findings allow us to conclude that Caspase-11 is being regulated by PGE₂, and from the other in vivo experiments that Caspase-11 is required for the allergic asthma phenotype. These data together therefore, and in combination with our in vitro data showing that PGE₂ blocks Caspase-11 induction via inhibition of Type I interferon production and Caspase-11 phosphorylation to limit activity, we feel our overall conclusion that Caspase-11 is required for allergic asthma and is a target for the protective effect of PGE₂ is well supported.

We therefore ask the reviewer to approve our manuscript for publication as we are confident that our work will be of interest and provoke further debate on the importance of Caspase 4/11 for disease.

Reviewers' comments:

Reviewer #4 (Remarks to the Author):

Caspase-4/11 can directly sense LPS. It is able to cleave the pore-forming protein GSDMD to induce pyroptotic cells, which contribute to anti-pathogen innate immune responses. Pyroptosis induces inflammatory diseases under certain circumstances. Caspase-11 activation is regulated by two steps, signal 1 (often called priming) for its expression and signal 2 for activation by LPS-induced oligomerization. In the submitted manuscript, Zaslona et al showed that PGE2 can suppress both caspase-4/11 expression and activation in LPS-stimulated cell death. Their biochemical analysis showed that TLR4-induced IFN β , which is required for signal 1, is attenuated in response to PGE2 treatment. The data are reasonable, but are predicted as a number of papers have already reported the suppressive impact of PGE2 treatment on TLR4 signaling (Nat Immunol. 2018 Dec;19(12):1309-1318., J Immunol. 2008 Feb 15;180(4):2125-31). Additionally, in the submitted manuscript, the Casp11 KO mice were observed to exhibit protection from asthma pathogenesis in a OVA model. Although this is an intriguing phenomenon, the contribution of PGE2 to asthma pathogenesis is controversial. A recent study with mice lacking EP2 (a PGE2 receptor, Sci Rep. 2016 Feb 8;6:20505) showed a positive role for PGE2 in asthma pathogenesis, which contradicts the authors' model, significantly undermining the value of their findings.

Major points.

In Fig. 1, 2, and 3, the authors showed inhibition of caspase-11 expression in TLR4-activated cells. This is not surprising as several papers have already reported the suppressive impact of PGE2 treatment on TLR4 signaling (Nat Immunol. 2018 Dec;19(12):1309-1318., J Immunol. 2008 Feb 15;180(4):2125-31).

Although caspase-4 expression is claimed to be attenuated by PGE2 treatment, the attenuation and induction by LPS (Fig. 1F) is not convincing. Other more quantitative and reproducible validations for caspase-4 expression, such as quantitative PCR, is required to make the conclusion.

PGE2 was shown to phosphorylate caspase-11 at S350; however, the physiological role of this phosphorylation was poorly characterized using an artificial over-expression setting. The difference between controls and S350A mutant are subtle (Fig. 4 F and S1). In addition, S350A does not seem to be conserved in human caspase-4 and other species, denying the critical role of S350 phosphorylation.

Although the authors assert a protective role for PGE2 in asthma patients, a numbers of studies have shown that PGE2 drives and sometimes has the opposite effect in asthma development (Mediators Inflamm. 2012;2012:645383). In the submitted Fig. 1D, EP2 (a PGE2 receptor) plays a dominant role in suppressing caspase-11 expression; however, EP2 KO mice were shown to be resistant to OVA-induced asthma (Sci Rep. 2016 Feb 8;6:20505.), which contradicts the authors' claim.

In methods section, it is not clear if indeed littermate controls were used in the asthma model. This should be clarified in the main text and legend. All Th responses seem to be attenuated in their Casp11 KO, raising a concern of off-target mutation in a regulator of T cell function. Insufficient backcross is another possibility. Littermate controls can exclude these concerns (Proc Natl Acad Sci U S A. 2015 Mar 10;112(10):3056-61.). The value of non littermate control study is highly limited.

Reviewer #4 (Remarks to the Author):

Caspase-4/11 can directly sense LPS. It is able to cleave the pore-forming protein GSDMD to induce pyroptotic cells, which contribute to anti-pathogen innate immune responses. Pyroptosis induces inflammatory diseases under certain circumstances. Caspase-11 activation is regulated by two steps, signal 1 (often called priming) for its expression and signal 2 for activation by LPS-induced oligomerization. In the submitted manuscript, Zaslona et al showed that PGE2 can suppress both caspase-4/11 expression and activation in LPS-stimulated cell death. Their biochemical analysis showed that TLR4-induced IFNB, which is required for signal 1, is attenuated in response to PGE2 treatment. The data are reasonable, but are predicted as a number of papers have already reported the suppressive impact of PGE2 treatment on TLR4 signaling (Nat Immunol. 2018 Dec;19(12):1309-1318., J Immunol. 2008 Feb 15;180(4):2125-31). Additionally, in the submitted manuscript, the Casp11 KO mice were observed to exhibit protection from asthma pathogenesis in a OVA model. Although this is an intriguing phenomenon, the contribution of PGE2 to asthma pathogenesis is controversial. A recent study with mice lacking EP2 (a PGE2 receptor, Sci Rep. 2016 Feb 8;6:20505) showed a positive role for PGE2 in asthma pathogenesis, which contradicts the authors' model, significantly undermining the value of their findings.

We agree that there are controversies regarding the contribution of PGE₂ to asthma pathogenesis and now we include the additional reference in the revised version of the manuscript. However, we have also included literature from various labs showing the suppressive role of PGE₂ and EP2/EP4/cAMP/PKA pathway both in asthma patients and asthma animal models. A suppressive role for PGE₂ in asthma is also validated in the well-known clinical phenomenon of Aspirin-Exacerbated Respiratory Disease, which represents over 10% of asthmatic patients (*Lancet*, 345 (1995), p. 1056, *N Engl J Med*. 2016 Feb 4;374(5):484-8.). We feel that our data supports the literature on PGE₂ being protective in asthma, but we are happy to provide the extra discussion on this issue.

Major points.

In Fig. 1, 2, and 3, the authors showed inhibition of caspase-11 expression in TLR4-activated cells. This is not surprising as several papers have already reported the suppressive impact of PGE2 treatment on TLR4 signaling (Nat Immunol. 2018 Dec;19(12):1309-1318., J Immunol. 2008 Feb 15;180(4):2125-31).

We agree that inhibition of TLR4 signaling by PGE₂ is well known. Our study however is the first to determine the inhibition of Caspase-4/11 expression and activity by PGE₂, which we feel is worth reporting in the context of our overall study.

Although caspase-4 expression is claimed to be attenuated by PGE2 treatment, the attenuation and induction by LPS (Fig. 1F) is not convincing. Other more quantitative and reproducible validations for caspase-4 expression, such as quantitative PCR, is required to make the conclusion.

As the reviewer requested we have run qPCR analysis on human PBMCs and demonstrated that PGE₂ inhibits LPS-induced caspase-4 expression as shown below. We have included this data in the new Fig. 1G and describe the data on page 5 of the revised manuscript.

Human PBMCs were pretreated with 1 μM PGE₂ for 30 min and stimulated with 100 ng/ml of LPS for 4 h. Cell lysates were subjected to qPCR analysis.

PGE₂ was shown to phosphorylate caspase-11 at S350; however, the physiological role of this phosphorylation was poorly characterized using an artificial over-expression setting. The difference between controls and S350A mutant are subtle (Fig. 4 F and S1). In addition, S350A does not seem to be conserved in human caspase-4 and other species, denying the critical role of S350 phosphorylation.

We feel that the difference between control and S350 is significant. Technical limitations did not allow us to study mutated caspase-11 proteins in physiological conditions. The relevant serine in human Caspase-4 is at position 366 in a PKA motif. We speculate it is equivalent to S350 in caspase-11. We now included 3D modeling using the PHYRE2 program and displayed using EzMol modelling software (displayed below) that the PKA motif in which S366 is located is in the same in IDR region as S350 in Caspase-11. We have included this model in Fig. 4H and described the data on page 9 of the revised manuscript.

Caspase-4 structure was generated using PHYRE2 protein modelling online tool and displayed utilizing EzMol. The position of S366 on Caspase-4 is shown.

Although the authors assert a protective role for PGE₂ in asthma patients, a numbers of studies have shown that PGE₂ drives and sometimes has the opposite effect in asthma development (Mediators Inflamm. 2012;2012:645383). In the submitted Fig. 1D, EP2 (a PGE₂ receptor) plays a dominant role in suppressing caspase-11 expression; however, EP2 KO mice were shown to be resistant to OVA-induced asthma (Sci Rep. 2016 Feb 8;6:20505.), which contradicts the authors' claim.

As mentioned above we are happy to discuss this controversy and now on page 12 of the revised manuscript include the citation on EP2 KO mice being resistant to asthma. Although we agree that the ability of PGE₂ to induce IgE is well documented, in the paper cited by the reviewer (*Mediators Inflamm. 2012;2012:645383*) PGE₂ is overall protective in asthma. We have also cited this paper on page 12 of the revised manuscript.

In methods section, it is not clear if indeed littermate controls were used in the asthma model. This should be clarified in the main text and legend. All Th responses seem to be attenuated in their Casp11 KO, raising a concern of off-target mutation in a regulator of T cell function. Insufficient backcross is another possibility. Littermate controls can exclude these concerns (Proc Natl Acad Sci U S A. 2015 Mar 10;112(10):3056-61.). The value of non littermate control study is highly limited.

We have used littermate controls in this study. To make this point clear we have added the following information to the Methods section:

Casp11^{-/-} mice on the C57BL/6J background were obtained from J. Yuan's laboratory (Harvard Medical School, USA) and were subsequently backcrossed onto the C57BL/6J (Harlan Laboratories, UK) background for another eight generations. Heterozygous

breeding pairs were used to generate wild type (WT) and Casp11^{-/-} littermates, which were used for all experiment described. Experiments were performed with 8-to 12-wk-old female mice bred under specific pathogen-free conditions, under license and approval of the local animal research ethics committee.

For genotyping of the offspring we have used a previously described method: *J Exp Med.* 2001 Jan 1;193(1):111-22. Briefly, PCR was used to genotype caspase-11 knockout mice. Genomic DNA was isolated from tails. The primer sequences used for the PCR were as follows:

SY-21, 5'-GGCATGGAGTCAGAGATGAAAGAC-3';

SY-22, 5'-GCCCATGTGGCATTACCTGCCAGC-3';

SYKO, 5'-AGATCTACACCTCTGCACAACCTGGGGT-3';

and PJK, 5'-TGGCGCTACCGGTGGATGTGGAATGTG-3'.

The wild-type genome of *caspase-11* could be detected using SY-21 and SY-22 (an ~200-bp PCR product), and the mutant caspase-11 gene could be detected using SYKO and PJK (an ~600-bp PCR product).

Reviewers' comments:

Reviewer #4 (Remarks to the Author):

In response to our comments, Zaslona et al. revised the manuscript with some additional data. The revision has slightly improved the manuscript; however, the major concerns were not addressed and remain unanswered. The new data provided is of poor quality. Suppression of TLR4 signaling by PGE2 is already well-known. The data provided disturbingly contradicts their hypothetical model of asthma. The roles of PGE2 in human asthma pathology and mouse OVA asthma model are controversial. Hence, I cannot recommend this for publication.

Major points

1. Suppression of TLR4 signaling by PGE2 (Fig. 1- 3), as the authors agreed in a point-by-point response to reviewers, is already well-known and has been reported by multiple groups, thus limiting any novelty.
2. New qPCR data (Fig. 1G in revised): this new data is missing replicate samples, and the reproducibility in repeat studies is questionable. The strong induction of CASP4 mRNA (over 100 fold) by LPS does not match with Fig. 1F western blot data. Quite disturbingly, the graph data in the point-by-point response looks different from the new Fig. 1G. PGE2 increased CASP4 expression in non-LPS treated cells (graph data in point-by-point response and Fig. 1F), which opposes their model.
3. Role of CASP11 phosphorylation at S350: this continues to be elusive and relies on an artificial overexpression study (Fig. 4E, F). This can be easily addressed in a more physiological setting such as reconstitution with mutants; however, this revised manuscript is still missing the doable study.
4. CASP4 phosphorylation: in the revised manuscript, a different residue (S366) is said to be functionally equivalent to S350 in CASP11; however, the location seems to be totally different (new Fig. 4, G, H) and the functional validation of CASP4 S366 mutants is missing.
5. Protection of EP2 KO mice from Asthma (Scientific Reports 2016): as the authors agreed, this previous study obviously opposes the authors' hypothesis. Though they claim PGE2 contextually plays an opposing role in the revised manuscript, a similar OVA model was employed in both studies. In addition, the context dependent, controversial phenomenon significantly undermines the value of story.

Reviewers' comments:

Reviewer #4 (Remarks to the Author):

In response to our comments, Zaslona et al. revised the manuscript with some additional data. The revision has slightly improved the manuscript; however, the major concerns were not addressed and remain unanswered. The new data provided is of poor quality. Suppression of TLR4 signaling by PGE2 is already well-known. The data provided disturbingly contradicts their hypothetical model of asthma. The roles of PGE2 in human asthma pathology and mouse OVA asthma model are controversial.

Response: we again maintain that inhibition by PGE2 of caspase-11 induction is a novel observation. Again we emphasize that multiple studies report the well-known phenomenon of Aspirin-Induced Asthma and the protective role of prostaglandins. The same stands for animal allergy/asthma models. Even the review used by the reviewer to state his case clearly speaks against his claims questioning his expertise. This is from the review the reviewer asked us to cite:

(Mediators Inflamm. 2012;2012:645383):

“PGE2 is commonly presumed to be a proinflammatory mediator and has been implicated in several inflammatory disease conditions, including rheumatoid arthritis [35]; however, PGE2 has protective effects in different organs, and respiratory system appears to be one of them in that PGE2 has beneficial effects [26, 36–38].

During the 1970s, PGE2 was shown to protect against bronchoconstriction produced by ultrasonically nebulized distilled water [39] and exercise-induced asthma [40]. In the early 1990s, Pavord et al. [41] showed that inhaled PGE2 protected against bronchial hyperreactivity to sodium metabisulphite in which bronchoconstriction is thought to be neurally mediated. In this study, furosemide protected against bronchoconstrictor challenges in asthma, and this effect may be mediated through PGE2.

Multiple subsequent studies have observed the bronchodilator effect of PGE2 in normal subjects [42] and patients with asthma and chronic bronchitis [43], showing that PGE2 attenuates bronchoconstriction, possibly inhibiting the release of the bronchoconstrictor mediators which are responsible for exercise bronchoconstriction. This prostanoid inhibits early and late allergen-induced bronchoconstriction, increasing the relaxation of airway smooth muscle and inhibiting the release of mast-cell mediators and the recruitment of inflammatory cells [34]. Moreover, PGE2 also decreases or inhibits the accompanying bronchial hyperresponsiveness to methacholine [36].

All these positive effects of PGE2 are mainly mediated through EP2 and EP4 receptors [44]. PGE2 can mediate bronchodilation via the EP2 receptor [45] and also anti-inflammatory effects via the EP2 and/or the EP4 receptor [46]. Also, EP2 plays an important role in aspirin-intolerant asthma because a reduction in released PGE2 and lower expression of its EP2 receptor provoked an increase in inflammatory process in the airways of these patients [47].”

The reviewer clearly doesn't understand the review he asked us to cite and then stated:

‘Hence, I cannot recommend this for publication. ‘

As regards the major points raised:

1. Suppression of TLR4 signaling by PGE2 (Fig. 1- 3), as the authors agreed in a point-by-point response to reviewers, is already well-known and has been reported by multiple groups, thus limiting any novelty.

We have replied to this point in each revision. None of previous studies examined Caspase-11/4 expression, so we again emphasize the novelty.

2. New qPCR data (Fig. 1G in revised): this new data is missing replicate samples, and the reproducibility in repeat studies is questionable. The strong induction of CASP4 mRNA (over 100 fold) by LPS does not match with Fig. 1F western blot data. Quite disturbingly, the graph data in the point-by-point response looks different from the new Fig. 1G. PGE2 increased CASP4 expression in non-LPS treated cells (graph data in point-by-point response and Fig. 1F), which opposes their model.

Response: these data are to complement our extensive data in the murine system and are from one human subject, hence the lack of statistical analysis. The graph data in the point-by-point response is identical to Fig 1G but included a control histogram, which is set at '1' in Figure 1G in the paper. PGE2 only marginally increased CASP4 expression in non-LPS treated cells (1.4 fold increase) which is much less than LPS (4.5 fold). Also, in Fig 1G, we measure the induction of transcription of Caspase-4 as instructed by the reviewer and PGE2 has no effect on that response.

3. Role of CASP11 phosphorylation at S350: this continues to be elusive and relies on an artificial overexpression study (Fig. 4E, F). This can be easily addressed in a more physiological setting such as reconstitution with mutants; however, this revised manuscript is still missing the doable study.

Response: we agreed to remove phosphorylation data from the manuscript

4. CASP4 phosphorylation: in the revised manuscript, a different residue (S366) is said to be functionally equivalent to S350 in CASP11; however, the location seems to be totally different (new Fig. 4, G, H) and the functional validation of CASP4 S366 mutants is missing.

Response: we agreed to remove phosphorylation data from the manuscript

5. Protection of EP2 KO mice from Asthma (Scientific Reports 2016): as the authors agreed, this previous study obviously opposes the authors' hypothesis. Though they claim PGE2 contextually plays an opposing role in the revised manuscript, a similar OVA model was employed in both studies. In addition, the context dependent, controversial phenomenon significantly undermines the value of story.

Response: again a puzzling comment. It is not unusual for studies to disagree and we feel the community should decide which experiment is likely to be correct given the totality of our data. Again we emphasize that the paper mentioned is in a minority, with most papers including ours showing an inhibitory function of PGE2-EP2 signaling in allergy/asthma.

REVIEWERS' COMMENTS:

Reviewer #3 (Remarks to the Author):

Fig. 1A, the effect of PGE2 inhibiting LPS transfection-induced pyroptosis is quite weak (from 55% to 40% on a background of >20%). In Fig. 1B, PGE2 does not inhibit the basal level of caspase-11 transcription and only partially inhibits LPS-induced caspase-11 transcription. These effects are indeed subtle. I think that the authors should do a dose titration and see if a strong inhibition can be observed with such a potent pharmacological agent. Also, does PGE2 inhibit caspase-1 expression? This is an excellent and easy control for Fig. 1C data. The authors just need to reblot that gel with different antibodies.

Fig. 1D shows that EP2 receptor-deficient BMDMs had a much higher basal expression of caspase-11. The authors wrote that "The preliminary data show that EP2 receptor-deficient BMDMs had increased expression of caspase-11, suggesting that endogenous PGE2 – EP2 signalling is an inhibitory signal on caspase-11 expression in resting macrophages (Fig. 1D)." I am not sure as whether preliminary data should be published in a peer-reviewed journal. Are the knockout macrophages more sensitive to LPS transfection-induced pyroptosis? Does PGE2 still show an inhibitory effect in the knockout BMDMs? Without further elaboration on the knockout BMDMs, this single piece of WB has little meaning.

In Fig. 2A, it appears that PGE has a dramatic effect of inhibition LPS-induced IFN β transcription. But why the inhibition on caspase-11 expression is weak in Fig. 1? Is it because at the protein level IFN β production does not drop that much in the presence of PGE2. The authors should do an ELISA assay of IFN β production to investigate whether this is the case, which will better clarify their model.

Again, Fig. 2D is a preliminary data. I guess peer reviewed journals only publish solid and conclusive data.

Data in Fig. 3 seem to challenge the hypothesis that PGE2 inhibits LPS-induced pyroptosis through inhibiting caspase-11 expression, which were suggested by results in Fig. 1 and 2. This presentation confuses the audiences what is going on here with potential mechanism of action of PGE2.

The asthma model in mice does not reveal whether the effect of PGE2 is due to inhibited caspase-11 expression or not. Indeed, the authors indicated in their model that PGE2 may also inhibit gasdermin D activation by caspase-11, but provided no data on that. It is quite easy and straightforward that the authors can block gasdermin D cleavage both in BMDMs and asthma to show whether this is the case. If so, this will help to clarify the puzzling data in Fig. 3 suggesting the mechanism of action of PGE2 is not due to inhibition of caspase-11 expression.

There were several places where referring to panels in Fig. 4 is wrong in the text description.

Response to Reviewers

Reviewer #3 (Remarks to the Author):

Fig. 1A, the effect of PGE2 inhibiting LPS transfection-induced pyroptosis is quite weak (from 55% to 40% on a background of >20%). In Fig. 1B, PGE2 does not inhibit the basal level of caspase-11 transcription and only partially inhibits LPS-induced caspase-11 transcription. These effects are indeed subtle. I think that the authors should do a dose titration and see if a strong inhibition can be observed with such a potent pharmacological agent. Also, does PGE2 inhibit caspase-1 expression? This is an excellent and easy control for Fig. 1C data. The authors just need to reblot that gel with different antibodies.

We are of the opinion that the effect here is significant and taking all our data together we are confident to conclude that PGE2 blocks caspase-11 expression via inhibition of IFN-beta production and also caspase-11 activation. Our study focuses on caspase-11 and the effect on caspase-1 has been studied previously (*J Immunol.* 2015 Jun 1;194(11):5472-5487. doi: 10.4049/jimmunol.1401343) and is beyond the scope of this manuscript.

Fig. 1D shows that EP2 receptor-deficient BMDMs had a much higher basal expression of caspase-11. The authors wrote that "The preliminary data show that EP2 receptor-deficient BMDMs had increased expression of caspase-11, suggesting that endogenous PGE2 – EP2 signalling is an inhibitory signal on caspase-11 expression in resting macrophages (Fig. 1D)." I am not sure as whether preliminary data should be published in a peer-reviewed journal. Are the knockout macrophages more sensitive to LPS transfection-induced pyroptosis? Does PGE2 still show an inhibitory effect in the knockout BMDMs? Without further elaboration on the knockout BMDMs, this single piece of WB has little meaning.

We agree that this is preliminary data and so it is stated as such in the manuscript. However, we feel it is still worth reporting since EP2 and EP4 are the most predominant receptor on macrophages and mediate most inhibitory effects of PGE2. (*Nat Med.* 2009 Jan;15(1):42-9. doi: 10.1038/nm.1905. Epub 2008 Nov 21.; *J Immunol.* 2008 Feb 15;180(4):2125-31).

In Fig. 2A, it appears that PGE has a dramatic effect of inhibition LPS-induced IFN β transcription. But why the inhibition on caspase-11 expression is weak in Fig. 1? Is it because at the protein level IFN β production does not drop that much in the presence of PGE2. The authors should do an ELISA assay of IFN β production to investigate whether this is the case, which will better clarify their model.

The effect of PGE2 on IFN-beta protein production has been previously reported (*J Immunol.* 2008 Feb 15;180(4):2125-31.) As above we are of the opinion that the effect on caspase-11 is significant - it could well be that there is residual IFN-beta protein causing some induction of caspase-11 but the effect on transcription of caspase-11 is clear.

Again, Fig. 2D is a preliminary data. I guess peer reviewed journals only publish solid and conclusive data.

The fact that LPS-induced caspase-11 transcription is driven by STAT1 is well known and the exact caspase-11 promoter region used in Fig. 2D to generate oligopulldown has been reported previously (*J Biol Chem.* 2002 Nov 1;277(44):41624-30. Epub 2002 Aug 26).

Caspase-11 gene expression in response to lipopolysaccharide and interferon-gamma requires nuclear factor-kappa B and signal transducer and activator of transcription (STAT) 1).

We have performed experiments showing inhibition of caspase-11 and IFNbeta transcripts and STAT3 phosphorylation by PGE2 at least 3 independent times using multiple mice. The experiment in Fig.2D, was an additional approach that provided consistent results with the aforementioned experiments, and did not require 3 independent repeats, which we felt would be a waste of mice, time and money. We have labelled it as a preliminary data.

Data in Fig. 3 seem to challenge the hypothesis that PGE2 inhibits LPS-induced pyroptosis through inhibiting caspase-11 expression, which were suggested by results in Fig. 1 and 2. This presentation confuses the audiences what is going on here with potential mechanism of action of PGE2.

PGE2 inhibits pyroptosis by inhibition of caspase-11 expression, however it can also inhibit pyroptosis independent of inhibition of caspase-11 expression, when given after LPS priming, most probably by inhibition of caspase-11 activity, by a mechanism which requires further investigation. This is discussed in the manuscript.

The asthma model in mice does not reveal whether the effect of PGE2 is due to inhibited caspase-11 expression or not. Indeed, the authors indicated in their model that PGE2 may also inhibit gasdermin D activation by caspase-11, but provided no data on that. It is quite easy and straightforward that the authors can block gasdermin D cleavage both in BMDMs and asthma to show whether this is the case. If so, this will help to clarify the puzzling data in Fig. 3 suggesting the mechanism of action of PGE2 is not due to inhibition of caspase-11 expression.

Gasdermin D may well be involved here but experiments on this are beyond the scope of the current manuscript. Figure 3 shows that PGE2 can also inhibit pyroptosis when given after LPS priming without affecting caspase-11 protein expression, which is likely to be an effect on caspase-11 activity. This is discussed in the manuscript.

There were several places where referring to panels in Fig. 4 is wrong in the text description.

We thank the reviewer for spotting that error which is now corrected.